

# Origin, size distribution and hygroscopic properties of marine aerosols in the south-western Indian Ocean: report of 6 campaigns of shipborne observations

Meredith Dournaux[1], Pierre Tulet[1], Joris Pianezze[1], Jérome Brioude[2], Jean-Marc Metzger[3], and Melilotus Thyssen[4]

[1]LAERO, Laboratoire d'Aérologie (UMR 5560 CNRS, UT3, IRD), Toulouse France
[2]LACy, Laboratoire de l'Atmosphère et des Cyclones (UMR 8105 CNRS, Université de la Réunion, Météo-France), Saint-Denis de la Réunion, France
[3]OSU-R, Observatoire des Sciences de l'Univers de La Réunion (UAR 3365, CNRS, Université de la Réunion, Météo-France), Saint-Denis de la Réunion, France
[4]MIO, Institut Méditerranéen d'Océanologie (UMR 235 Aix Marseille Univ, Université de Toulon, CNRS, IRD), Marseille, France

**Correspondence:** Pierre Tulet (pierre.tulet@cnrs.fr)

**Abstract.** Marine aerosol observations from 6 shipborne campaigns carried out in 2021 and 2023 in the southwest Indian Ocean are presented. A set of aerosol instruments is used to study the spatial and temporal variability of the aerosol size distribution, cloud condensation nuclei (CCN), activation diameters, and hygroscopicity ($\kappa$). Total number of aerosols (Na) shows concentration above $1500\,\mathrm{cm^{-3}}$ in polluted areas, and between $100\text{-}1500\,\mathrm{cm^{-3}}$ in the open ocean. CCN measurements

(0.2 %, 0.4 % supersaturation) range from 40 to $500\,\mathrm{cm^{-3}}$. At 0.2 % (0.4 %) supersaturation, the average activation diameter is 104 (76) nm and $\kappa$ is 0.36 (0.25). Using a back-trajectory model, the aerosol data were classified into three source regions. Aerosols are hydrophobic in the continental group ($\kappa$ from 0.1 to 0.13), hydrophilic in the Subtropical Indian Ocean group ($\kappa$ from 0.24 to 0.4), and intermediate values are found in the Southern Indian Ocean group ($\kappa$ from 0.17 to 0.22). Subtropical Indian Ocean $\kappa$ increases with stronger wind speeds. Southern Indian Ocean $\kappa$ decreases significantly (between 0.2 % and 0.4 %

supersaturation) with stronger wind speeds, probably due to a higher concentration of organic species on the smallest particle surface. High aerosol concentration events are presented. Pollution related to air masses passing through a well-developed continental boundary layer. Nucleation triggered by clear skies between two cloudy periods. Arrival of air masses at the ship's location after a precipitation event.

## 1 Introduction

Aerosols have been identified as playing a key role in climate, cloud formation, and cloud lifetime through their direct and indirect effect (IPCC, 2013, 2021; Wall et al., 2023). Among them, marine aerosols constitute a significant mass proportion of particles with global emissions estimated between 2,000 to 10,000 Tg $\mathrm{year^{-1}}$ (O'Dowd et al., 1997; Bates et al., 2005; Textor et al., 2006; de Leeuw et al., 2011). Some come from local sources and are either mechanically injected into the atmosphere via bubble bursting (Lewis and Schwartz, 2004; de Leeuw et al., 2011), wave crest tearing (Monahan et al., 1986), or chemically





formed by gas to particle conversion after gas emission from the ocean essentially due to phytoplankton activity (Saltzman, 2009; de Leeuw et al., 2014). Others come from remote sources and are transported from landmasses towards the ocean as dust (Schulz et al., 2012), biomass burning aerosols, or particles originating from fossil fuel combustion (Ramanathan et al., 2001; Novakov et al., 2000). This diversity of origins makes the size and chemical composition of marine aerosols highly variable. These two characteristics are essential in determining the aerosol hygroscopicity, which is the ability of aerosols to take up

the surrounding atmospheric water vapor and grow to larger sizes. Thus, according to their hygroscopicity, marine aerosols may act as cloud condensation nuclei (CCN) and activate as cloud droplets leading to cloud formation (Köhler, 1936). Clouds in turn affect the Earth's energy balance by reflecting short-wave radiation and absorbing and emitting long-wave radiation. Indeed, a change in aerosol concentration or its properties affects cloud droplet number concentration (Twomey, 1974, 1977) and cloud lifetime leading to changes in precipitation (Albrecht, 1989). For instance, low clouds as marine stratocumulus,

which is the most widely spread cloud type on Earth (Warren et al., 1988, 2007), has been intensively studied due to their strong sensitivity to aerosol concentration changes which affect their cloud droplet number (Stevens et al., 1998; Sandu et al., 2008; Brioude et al., 2009; Jia et al., 2019). Therefore, the complex interactions between aerosols and these clouds in the marine environment constitute one of the largest uncertainties in climate models (Carslaw et al., 2013; Simpkins, 2018) and contribute to less accurate climate predictions. One of the primary reasons for these uncertainties lie in the inadequate depiction

of aerosol sources in the remote marine ocean (Carslaw et al., 2017). In this region where aerosol loads are lowest, cloud droplet sensitivities are greatest (Moore et al., 2013).

To improve our understanding of the life cycle, size distribution, and chemical composition of marine aerosols and their impact on climate; several studies were conducted mainly in the Atlantic, the Pacific, and the Northern Indian oceans (Heintzenberg et al., 2000). Some studies focused on the physical properties of marine aerosols. Flores et al. (2020) provided recent

marine aerosol total number concentration and size distribution measurements along coastal and remote areas of the Pacific and the Atlantic oceans during the Tara Pacific Expedition (2016-2018). In the open ocean and far from continental sources, the average Na (0.03-10 μm) was $180 \pm 51 \ \text{cm}^{-3}$ and $864 \pm 806 \ \text{cm}^{-3}$ in the Pacific and Atlantic Oceans respectively. The associated size distributions were bimodal with an Aitken mode median diameter between 30-50 nm and an accumulation mode median diameter around 200 nm. Under continental influence, Na exceeded $5000 \ \text{cm}^{-3}$ in the Atlantic Ocean and 2718

$\pm 1560 \ \text{cm}^{-3}$ on average in the western Pacific Ocean. The associated size distributions were monomodal with a mean diameter around 100 nm. In the Indian Ocean, Pant et al. (2009) measured Na and its size distribution (16-700 nm and 0.5-20 μm in diameter) from 14° N to 56° S (January-March 2004). They reported a minimum concentration of micrometer aerosols in the trade wind region around 11° S and a bimodal size distribution with an Aitken mode mean diameter around 50 nm in addition to an accumulation mode mean diameter around 130 nm. North and south of the trade wind region, the concentration

of micrometer particles was higher, and the maximum concentration ($5.8 \pm 3.5 \ \text{cm}^{-3}$) was recorded in the roaring forties from 40°-56° S. South of the trade wind region the aerosol number size distribution was more monomodal with a higher number concentration of aerosols in the Aitken mode. Other studies focused on the chemical composition of aerosols. In the Northern Atlantic Ocean and along the Northern American shores, Kasparian et al. (2017) showed that the organic fraction of aerosols was higher (reaching up to 10 % of the aerosol composition between 30-32 PSU and 12-14°C) over colder and less salty wa-



ters outside of the Gulf Stream compared with warm and salty waters within the Gulf Stream. Yoon et al. (2007) investigated

seasonal chemical composition of marine aerosols in the North Atlantic Ocean and found a maximum (minimum) mass con-

centration of sea salt in the supermicron mode during winter (summer) due to stronger wind speeds during this season. On the

contrary, non sea-salt sulfate concentration in the submicron mode exhibited higher concentration during summer than winter.

Several studies conducted both in the field or in laboratory (O'Dowd and de Leeuw, 2007; Christiansen et al., 2019) have been

dedicated to the physical mechanisms generating sea spray aerosols. Other studies focused on CCN number concentration. In

the Southern Ocean, Quinn et al. (2017) identified a large portion of Aitken mode particles acting as CCN at SS > 0.5 %. In the

Southern Ocean, Tatzelt et al. (2022) reported shipborne CCN number concentrations ranging between 3-590 $\mathrm{cm}^{-3}$ at 0.3 %

SS in the austral summer during cruises conducted between the southern tips of Argentina, South Africa, and Australia. At the

same SS, Sanchez et al. (2021) reported airborne CCN measurements conducted in the marine boundary layer (MBL) ranging

from 17 to 264 $\mathrm{cm}^{-3}$ between Tasmania and 62° S in the austral summer. Marine aerosol hygroscopicity (materialized by the

Kappa-Köhler parameter, $\kappa$) has been prescribed to a single value of $0.7 \pm 0.2$ (Andreae and Rosenfeld, 2008), or $0.72 \pm 0.24$

according to global model simulation (Pringle et al., 2010). However, several field campaigns reported a large variability of $\kappa$

values according to the influence of air masses and the presence of marine biologic activity. For instance, Huang et al. (2022)

studied the spatial variation of marine aerosol ($\leq$ 300 nm in diameter) $\kappa$ over 100° of latitude in the Atlantic ocean. They mea-

sured $\kappa$ values ranging from 0.14 to 0.16 in the equatorial region influenced by biomass burning emissions coming from Africa

and characterized by wind speeds smaller than 4 $\mathrm{m\ s}^{-1}$. In comparison, regions with air masses coming from the ocean and

characterized by strong wind speed ($\geq$ 10 $\mathrm{m\ s}^{-1}$) were associated with higher $\kappa$ values ranging from 0.86 to 1.06 for 300 nm

particles. Jung et al. (2013) reported $\kappa$ values as low as 0.02-0.03 for accumulation mode particles during an intense Saharian

dust episode arriving at Barbados. In the same area, but without the influence of Saharian dust, Wex et al. (2016) reported an

average $\kappa$ value of 0.66 for particles in the size range 50-200 nm, indicating the presence of sulfates generally formed during

new particle formation events. In the summer month at Barbados, Kristensen et al. (2016) reported $\kappa$ values in the range 0.2-0.5

for supersaturations between 0.1-0.7 %, and attributed these low values to the presence of a significant organic volume fraction

in the particles ranging between 50-200 nm, confirmed by transmission electron microscopy evaporation studies. Shipborne

measurements performed in the tropical Atlantic Ocean between Portugal and Cape Verde exhibited $\kappa$ between 1.15-1.4 for

marine air masses, $\kappa$ between 0.92-1.24 for African air masses, and $\kappa$ between 0.75-0.89 for continental air masses (Good

et al., 2010). Within the marine boundary layer west of the North American coast, Roberts et al. (2010) reported an average

$\kappa$ value of 0.21 at 0.35 % supersaturation for air masses that originated from the North Pacific Ocean. In the Southern Ocean,

between Tasmania and 62° S, $\kappa$ values exhibit a large variability ranging between 0-1.2 (Sanchez et al., 2021), according to

the processes impacting the air mass history (aging, recent particle formation and aging, scavenging, recent particle formation

and scavenging). In this study, $\kappa$ values also varied according to the latitudes, with higher values ($\kappa$    1) at lower latitudes due

to primary emissions, and lower values found at higher latitudes ($\kappa$ between 0.6-0.9 in presence of sulfate species and $\kappa < 0.2$

in presence of organic species from biogenic emissions). However, few studies attempted to link marine aerosol number con-

centration, size distribution, and hygroscopic properties in the Southern Indian Ocean and the Southern Ocean. Additionally,




most of the campaigns already carried out were short-term field campaigns, targeted specific remote or coastal regions, and
used different instrumentation from one expedition to another; leading to the generation of unmatched data sets.

The Marion Dufresne Atmospheric Program-Indian Ocean (MAP-IO) was launched in 2021 (www.mapio.re; (Tulet et al.,
2024)). The program relies on continuous atmospheric and oceanic measurements realized aboard the Marion Dufresne II
vessel (https://taaf.fr/collectivites/le-marion-dufresne/) over the Southern Indian Ocean. One of its objectives is to better char-
acterize the properties of marine aerosols (i.e. their number concentration, size distribution, and hygroscopic properties) in
a poorly documented area far from the main anthropogenic influences. One particularity of MAP-IO is that it uses the same
vessel and the same instrumentation over, to the authors knowledge, the longest sampling period and greatest spatial cov-
erage ever undertaken in this region. Each year the Marion Dufresne covers a large panel of latitudes extending from the
sub-equatorial region until the subantarctic front (20° S – 60° S). This offers various possibilities to study the impact of local
in-situ conditions on marine aerosol size distribution and hygroscopic properties. This includes weak to strong wind speeds,
calm to rough ocean, sunny to cloudy areas, regions of intense biological activity, pristine areas, and coastal zones influenced
by human activity. The large spatio-temporal coverage also allows us to capture large-scale influence via long-range transport
of aerosols over the Southern Indian Ocean. This diversity is translated in the great spatial and temporal variability of Na and
size distribution of marine aerosols observed during campaigns that took place between 2021 and 2023 and presented in Tulet
et al. (2024). Complementary measurements involving a photometer, an automated flow cytometer, and gas analyzers allow to
identify terrestrial transport and any relation with phytoplankton distribution and functional composition. The objective of this
paper is to present the variability of marine aerosol hygroscopicity and size distribution over the eastern and southern Indian
Ocean. Six oceanographic campaigns are used to provide the spatio-temporal distribution of marine aerosol size distribution
and their hygroscopicity from SMPS, OPC-N3, and CCN-100 analyzers. As aerosol hygroscopicity is closely linked to their
chemical composition which in turn depends on local and large-scale conditions, we propose to use co-located measurements
of wind speed, wave height, gases, phytoplankton functional groups distribution, and back trajectories simulations to analyze
the differences in the calculated hygroscopicity. This paper is organized as follows: section 2 is dedicated to the presentation
of the campaigns, and the in situ conditions encountered. Section 3 describes instrumentation and section 4 focuses on data
processing. Section 5 deals with the spatial and temporal variability of marine aerosols properties. Section 6 presents the origin
of analyzed air masses, section 7 focuses on particular events, and conclusions and perspectives are presented in section 8.

## 2   Campaigns overview and in situ conditions observed

### 2.1   Campaigns overview

Between January 2021 and March 2023, MAP-IO carried out a total of 16 campaigns aboard the Marion Dufresne, including
6 during which the aerosol instruments performed well and are presented in this article (Fig. 1, Table 1). During these 6
campaigns, the spatial coverage of the Marion Dufresne extended from latitudes -10.65° S to -60° S and longitudes 31° E to
83.2° E and thus encountered various environmental conditions (Fig. 1).





The SWINGS campaign (Fig. 1, blue track) took place during the austral summer from January 13th to March 8th. The aims were to collect CO2, pH, and fish population density measurements at different latitudes and depths in the Southern Ocean. During this campaign, the Marion Dufresne route begins along coastal regions passing south of Madagascar and southeast of Africa, then moves on to open ocean where it stops at Crozets then Kerguelen islands before returning to Reunion island. The

SCRATCH campaign (Fig. 1, red track) took place during austral winter and was carried out from July 1st to 22nd. The aim was to better characterize the region using a combination of biological and geological techniques. During this campaign, the vessel passed east of Madagascar and stayed in the northern part of the Mozambique Channel before going back to Reunion island. The MAYOBS campaign (Fig. 1, green track) was carried out during austral winter from September 13th to October 3rd, as part of the monitoring of an underwater volcanic eruption that began around Mayotte in 2018. The route of the vessel

was similar to the one during the SCRATCH campaign. The OP3 and OP4 campaigns (Fig. 1, orange and lime tracks) were carried out under TAAF charter during austral spring from October 28th to November 28th, and from November 28th to December 30th respectively. During these two campaigns, the Marion Dufresne took the same routes: Reunion island-Crozets islands-Kerguelen islands-Amsterdam island-Reunion island. The OBSAUSTRAL campaign (Fig. 1, purple track) took place during the austral summer in 2023 from January 18th to February 28th. The objectives were the same as the SWINGS cam-

paign, including the monitoring of seismic activity at ocean ridges and the identification of cetaceans vocal signature using hydrophones. The vessel route was similar to the one during OP3 and OP4 with one exception: it went further to the south between Crozets islands and Kerguelen islands and further to the east between Amsterdam island and Reunion island.

## 2.2  Atmospheric and oceanic conditions observed

Between 10° S and 25° S, westward surface atmospheric and oceanic circulation are prevailing with southeast equatorial winds

and the South Equatorial Current (SEC). Between Madagascar and 80° E during summer (Fig. 1 red arrows), surface winds are weaker and slightly oriented west to northwest. During winter, (Fig. 1 blue arrows) surface winds are stronger and oriented northwest ((Schott et al., 2009), and references therein). In this latitude range, the average wind speed measured onboard the Marion Dufresne is $8.9 \pm 4.6$ m s$^{-1}$, the average wave height is $4.5 \pm 2.3$ m, and the average nanophytoplankton abundance is $669.1 \pm 583.7$ cell mL$^{-1}$ (Fig. 2). Madagascar's high plateaus play a role in regional circulation by splitting the trade winds

into two branches. A northeasterly flow, which notably affected the region of the SCRATCH and MAYOBS campaigns. A southeasterly flow, which affected the beginning of the SWINGS campaign's route between Reunion Island and Madagascar. The oceanic circulation also divides into two branches at the northeastern tip of Madagascar with one bypassing the island to the north (North East Madagascar Current, NEMC) and the other to the south (South East Madagascar Current, SEMC). These two currents contribute to the development of the Mozambique Channel eddies west of Madagascar. Between the southern tip

of Madagascar and 30° S, the wind direction progressively changes and becomes eastward when encountering strong westerlies typically extending from 35° S to 60° S. The SEMC flows towards the southeast coast of Africa where it becomes the Agulhas Current (AC). Although the southeast coast of Madagascar is an oligotrophic region, phytoplankton blooms occur in January and April and can extend over 2500 km$^2$ eastward (Longhurst, 2001; Dilmahamod et al., 2019). However, during the sampled period, no enhancement of the phytoplankton abundance was observed in this region during the SWINGS campaign (Fig. 2c).





The area extending southeast of Madagascar has rather been considered as pristine in the literature (Fig. 1 blue shaded area) (Mallet et al., 2018). At 40° S, the AC is retroflected and goes eastward. Then it becomes the South Indian Current (SIC) and flows northeastward. The subantarctic front is located south of it, with the northern Subantarctic front typically between 45° S and 50° S and the southern Subantarctic front located between 50° S and 60° S. This current is characterized by a strong sea temperature and salinity gradient between the Subtropical zone and the Antarctic zone which marks the northern boundary

of the Southern Ocean (Giglio and Johnson, 2016). The Marion Dufresne crossed the subantarctic front during the SWINGS, OBSAUSTRAL, OP3, and OP4 campaigns. In this area, phytoplankton blooms driven by nitrate, phosphate, or iron water fertilization were observed during the austral summer on the Crozet Islands (46° S, 51° E) and Kerguelen Plateau (49° S, 69° S) (Sedwick et al., 2002; Blain et al., 2008). Measurements of nanophytoplankton abundance were performed during SWINGS and OBSAUSTRAL, during the austral summer, and showed a clear signal of enhancement between 40-55° S (Fig. 2c). The

polar front is further in the south between 55° S and 60° S and marks the boundary between the warmer subantarctic water and the cold Antarctic water. The Southern Ocean is the roughest ocean on Earth due to the absence of land (Young, 1999). Even in the austral summer, Derkani et al. (2021) measured average wind speeds of 11 m s$^{-1}$ and swells in excess of 3.5 m. During the SWINGS and OBSAUSTRAL campaigns, average winds also exceeded 10 m s$^{-1}$ and average wave heights exceeded 5 m. Three major storms were documented during the SWINGS campaign in 2021. The first one located south of

Crozets islands, the second one south of Kerguelen islands and the third one north of Kerguelen islands. The maximum wind speed was respectively 23 m s$^{-1}$, 33 m s$^{-1}$, and 27 m s$^{-1}$. The maximum wave height was respectively 14 m, 21 m, and 15 m. The nanophytoplankton abundance is lower (400-800 cell mL$^{-1}$) along the storm tracks due to a more important mixing which deepens the ocean mixing layer, and limits the light available for phytoplankton growth (Fragoso et al., 2024).

For more information on the climatology of the south-western Indian Ocean and the Southern Ocean, see for example the

studies by Schott et al. (2009) or Mondal et al. (2022).



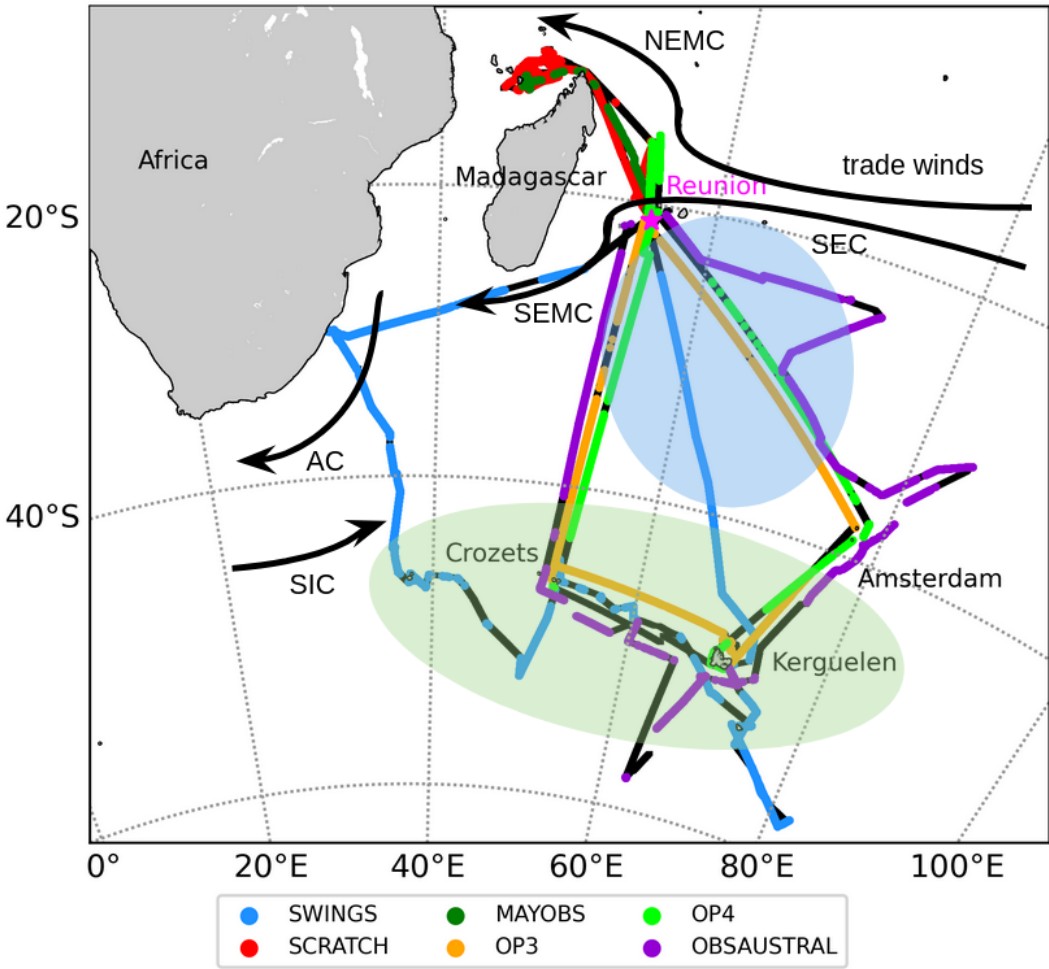

**Figure 1.** Marion Dufresne paths colored by campaign in 2021 and 2023 during four specific campaigns (SWINGS, SCRATCH and MAY-OBS 2021, OBS AUSTRAL 2023) and two port operations (OP3 and OP4 2021). Filtered data is colored according to the campaigns whereas black solid lines present the total path of the Marion Dufresne before filtering. Paths of OP3, OP4, and OBSAUSTRAL are shifted in longitude (between -1.4° and 1.2°) and latitude (between -1.6° and 1°) from their original location to give a better view of the different campaigns. Black arrows represent the surface atmospheric and surface oceanic circulation in the region. Green shaded area is the region where phytoplankton are the most abundant and are generally observed in the austral summer. Blue shaded area is the pristine region observed in the Southern Indian ocean.





| Name of campaign | Start to end date | Analyzed data |
|---|---|---|
| SWINGS | 2021-01-13 to 2021-03-08 | CPC, SMPS, OPC-N3, CCN-100 |
| SCRATCH | 2021-07-01 to 2021-07-22 | SMPS, OPC-N3, CCN-100 |
| MAYOBS | 2021-09-13 to 2021-10-03 | SMPS, OPC-N3, CCN-100 |
| OP3 | 2021-10-28 to 2021-11-28 | SMPS, OPC-N3, CCN-100 |
| OP4 | 2021-11-28 to 2021-12-30 | SMPS, OPC-N3, CCN-100 |
| OBSAUSTRAL | 2023-01-18 to 2023-02-28 | CPC, SMPS, OPC-N3, CCN-100 |

**Table 1.** Name and duration of the campaigns realized in 2021 and 2023 and list of data analyzed in the present paper.

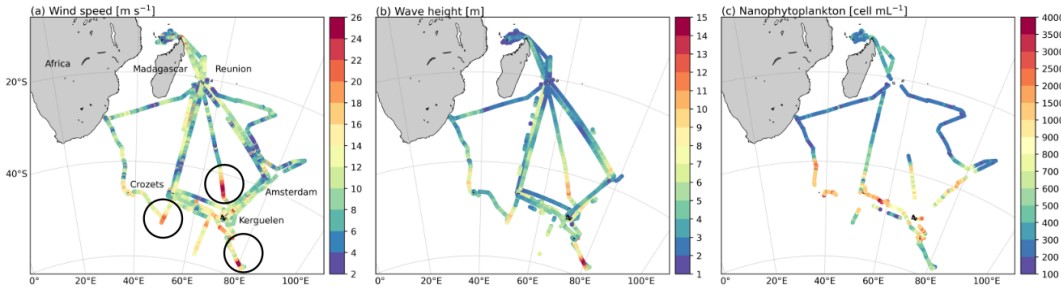

**Figure 2.** Marion Dufresne path colored by wind speed (a), wave height (b), and nanophytoplankton abundance (c) during the six campaigns analyzed. The three storms that occurred during SWINGS are circled in black.

# 3 Instrumentation description

In the framework of the MAP-IO program, the Marion Dufresne has been equipped with 19 measurement and remote sensing instruments described in Tulet et al. (2024). Among the 19 instruments, 7 are dedicated to the study of aerosols and atmospheric gas. In Figure 3, top left and right pictures present the aerosol and gas inlets located at the center of the vessel, upstream of the exhaust stack and 21 m above the sea level. The bottom picture of Figure 3 shows the aerosol and gas acquisition system and analyzers located in the meteorological laboratory and mounted on a shock-absorbing table.





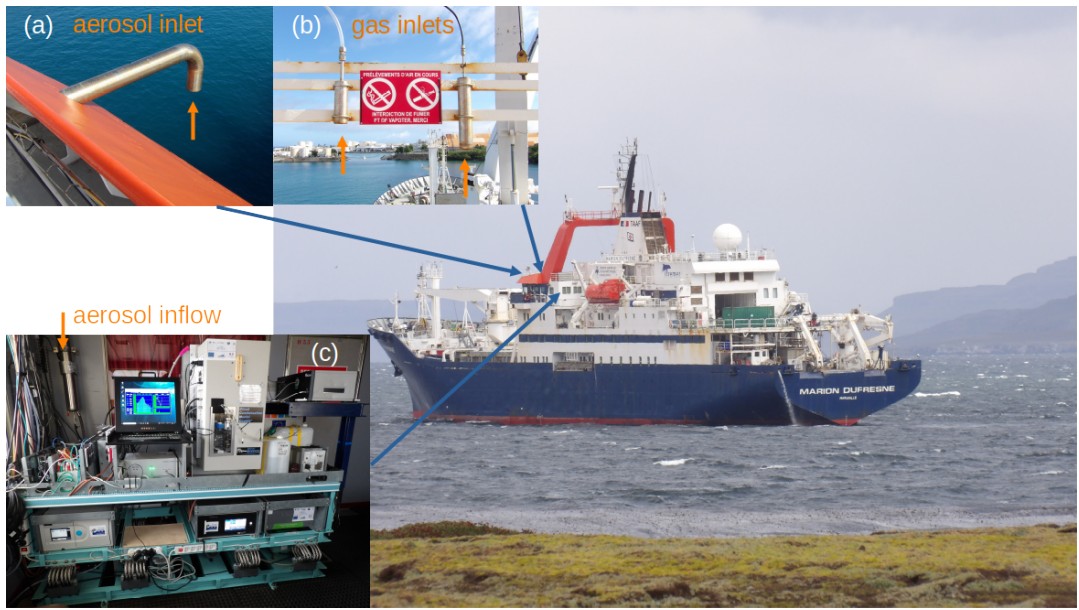

**Figure 3.** Picture of the Marion Dufresne in the Port-aux-Français Bay (February 2024). The aerosol (a) and gas inlets (b) are located above the wheelhouse. The acquisition and monitoring system (c) is located in the meteorological room, next to the wheelhouse and mounted above a shock-absorbing table. Aerosol monitoring systems are on the top shelf, right of the acquisition computer. Gas analyzers are on the bottom shelf. Photography credits: Meredith Dournaux.

Aerosol inflow enters the same sampling line which divides after a Nafion tube, used to dry the aerosol inflow, and direct them towards the different instruments. The distance between the inlets and the instruments (8 m) was carefully chosen to minimize aerosol losses along the sampling line. To minimize the loss of the largest particles (> 1 µm), three OPC-N3 are installed directly outside on the main deck. A Water-Based Condensation Particle Counter (CPC, model MAGICTM-200/210) measures particle number concentration within the size range 5 nm to 2.5 µm using a condensational growth system. A Scanning Mobility Particle Sizer (SMPS, model 4S) is used to measure the number size distribution of aerosols. It is composed of a Differential Mobility Particle Sizer (DMPS) and a CPC (model MAGICTM-200/210). The DMPS takes in the aerosol flow composed of dry particles and cloud droplets, the water of the latter is evaporated while penetrating the inlet. The aerosol population is first neutralized using an X-ray neutralizer (TSI, model) to provide the same electrical charge to all the particles, then selected using their size dependent electrical mobility and classified into 80 bins from 20 nm to 350 nm. The number of aerosols per bin is then determined by the CPC. The whole size range is scanned in five minutes.

To complete the aerosol size distribution towards super-micron diameters, three Optical Particle Counters (OPC-N3, model Alphasense) are used to measure the number concentration and the size of particles within the size range 0.35 µm to 40 µm. Using a calibration based on Mie theory, the OPC-N3 measures the light scattered by each particle of a sample air flow (sample flow rate of 210 mL $min^{-1}$) passing through a 658 nm laser beam. According to the intensity of the scattered light the particles size and concentration are determined, and permit the classification of each particle into one of the 24 bins covering





the measurement size range at a rate of about 10,000 particles per second. Generally, 100 % of the particles are detected at 0.35 μm and 50 % at 40 μm. (Alphasense User Manual, www.alphasense.com). In our case, as the OPC-N3 have no sampling line and are directly located outside, there is no loss of large aerosols except on the walls of the instruments. To study the aerosol activation properties a Cloud Condensation Nuclei Counter (CCN-100, DMT) is used to measure the number concentration of activated aerosols at different supersaturation (SS = 0.1, 0.2, 0.3, 0.4, 0.5, 0.6, 0.8, and 1.0 %), the two first set for fifteen minutes and the others for five minutes. Supersaturation is created by a three temperature control zones column mounted vertically. The temperature at the column wall increases gradually from the top to the bottom to create supersaturation. The aerosol sample enters at the top of the column and becomes progressively supersaturated with water vapor as it goes through the column. As the sample temperature is lower than the wall temperature and using the fact that water vapor diffusion in the air is faster than heat diffusion, water vapor condenses on particles to form droplets. Activated droplets are then counted by an internal OPC and distributed into 20 size bins going from 0.75 μm to 10 μm. Gas measurements are realized by a NOx analyzer (model Teledyne N500 CAPS), an O3 analyzer (model HORIBA APOA-370), and a Picarro analyzer (model Picarro G2401 CFKADS-2372) measuring atmospheric gas traces as CO, CO2, CH4, and H2O (ppb). The gas inlets are located right of the aerosol inlet (Fig. 2). Along with aerosol measurements, wind speed, and wind direction (m s$^{-1}$ and °), air temperature (°C), pressure (hPa), and humidity (%) are recorded by the Vaisala and Mercury meteorological stations with a sampling time step of 5 seconds and 1 minute respectively. Sea surface elevation (m) and ship position is also recorded by the inertial unit of the ship at a sampling time step of 1 second.

## 4   Data processing

### 4.1   Filtering steps

First the data is filtered to remove all the measurements likely to be contaminated by the exhaust of the vessel. Observations of time series of the total number concentration and the relative wind direction concluded that measurements realized within the direction range 90°-225° were associated with high aerosol concentrations (> 5000 cm$^{-3}$) lasting in time. A second filter combining relative low wind speed, vessel cruise speed, and gas concentration data was applied in an attempt to remove local episodes of pollution. Thus, all data collected at a wind speed smaller than 2 m s$^{-1}$ which may have inhibited pollution plume dilution around the inlets, were removed. In addition, data was removed when the ship was stationary, thus eliminating contamination by maintenance activities on board (painting, rust removal). To complete the filter, data with corresponding high concentrations of NO and NO2 (> 1 ppbv) were removed, the latter being relatively low (less than 1 ppbv) in remote marine environments. CO peaks of concentration were also identified following the semi-automatic method of detection described in El Yazidi et al. (2018). The use of different chemical tracers was not only useful when one of the analyzers did not work properly, but also allowed to confirm the good agreement between NOx and CO concentrations. A third step consisted in the quality control of the aerosol data. For SMPS data, the particle size distributions were filtered manually for each measurement to remove non physical variations (e.g. local pollution undetected by the dynamic and the chemical filters). The latter were noticeable by a brutal and short increase in the concentration of aerosols over the entire range of diameters known as "spikes."





SMPS data were quality controlled and corrected over the sampling periods using CPC data. For CCN-100 data, supersaturations of 0.2 % and 0.4 % were treated separately and only the data recorded on plateau (measurements realized approximately three minutes after a supersaturation change according to the CCN-100 manual guide) were averaged over five minutes to make them comparable to SMPS data.

The original raw SMPS dataset was made of 55144 measurements of aerosols which represents 192 days of measurements. After the first and second filtering steps, it consisted of 53201 measurements. After the manual filtering step, 36226 measurements are conserved which represents 66 % of the original dataset. In total, there were 29376 measurements taken in 2021 and 6850 measurements in 2023 as shown in Fig. 1. The start and end date of each campaign is resumed in Table 1. The data collected in 2022 were not used in this study due to SMPS maintenance and calibration.

## 4.2   Calculation of activation diameter and hygroscopicity parameter

Assuming that aerosols are internally mixed and that larger particles are activated preferentially before the smallest ones due to the curvature effect, we can determine the hygroscopicity of aerosols at the activation diameter by calculating activation diameters and deriving the hygroscopicity parameter at both supersaturation. Activation diameters were calculated using the total number of cloud condensation nuclei given by the CCN-100 and the number concentration of aerosols measured in each bin by the SMPS. The number of aerosols per bin were integrated from the largest diameter towards the smallest until matching the number of CCN. The activation diameter corresponds to this smallest diameter. Hygroscopicity parameter Kappa-Köhler was derived from the previously determined activation diameters as follows (Petters and Kreidenweis, 2007):

$$\kappa = \frac{4A^3}{27D_d^3 \ln^2 S_c} \tag{1}$$

$$A = \frac{4\sigma_{s/a} M_w}{RT \rho_w} \tag{2}$$

with A composed of $\sigma_s/a = 0.072$ J m$^{-2}$ the surface tension of pure water, $M_w = 0.018$ Kg mol$^{-1}$ the molecular weight of water, R = 8.314 J mol$^{-1}$ K$^{-1}$ the perfect gas constant, T the activation temperature in K and $\rho_w = 997$ kg m$^{-3}$ the density of water. $D_d$ is the activation diameter in nm calculated for a given supersaturation $S_c$. Note that hygroscopic particles have theoretical Kappa values between 0.5 and 1.4. Petters and Kreidenweis (2007) summarized several $\kappa$ values derived from CCN measurements. $\kappa$ values are 1.28 and 0.9 for NaCl and H$_2$SO$_4$ respectively. When the Na$^+$ ion is associated with other compounds, its $\kappa$ values decrease and range between 0.8 and 0.88. Ammonium sulfate has $\kappa$ values between 0.61 and 0.67. Organic compounds have $\kappa$ values between 0.01 and 0.4 and hydrophobic aerosols such as black carbon are 0.



## 4.3 Method of classification of air masses arriving along the Marion Dufresne path

Back-trajectories provide additional information to understand the origin of the air mass and thus help to better understand the observed aerosol size distribution and properties. For that, the FLEXPART version 10.4 has been used from the vessel's position.

The FLEXible PARTicle model is part of the multi-scale offline Lagrangian Particle Dispersion Models (LPDMs) which have been developed to simulate transport and turbulent mixing of aerosols and gasses in the atmosphere. The model can be run for forward or backward simulations. In backwards mode, the location where particles are released, called a "receptor," is defined within a longitude-latitude-altitude cell (Pisso et al., 2019). For this study, ERA-5 meteorological fields with a spatial resolution of 0.5° x 0.5° and time resolution of 1h were used as input. Five-day back trajectories were run each hour of campaign at the location of the vessel with a particle release between 200 and 250 m of altitude. To analyze the dataset according to the air mass origin, the area covered by the vessel was divided in three subdomains related to the main types of air masses. The continental air masses were identified as such when the residence time over the area north of 40° S and west of 50° E was the greatest. Air masses were classified in the Subtropical Indian Ocean group when the residence time over the area east of 50° E and north of 40° S was the greatest. Thus, the Southern Indian Ocean air masses were identified as such when the residence time over the area south of 40° S was the greatest. This classification led to 15385 receptors in the Southern Indian Ocean group (with spatial extension [20° S-60° S] [35° E to 80° E]), 5495 receptors in the Subtropical Indian Ocean group (with spatial extension [20° S-50° S] [55° E-80° E]), and 4048 receptors in the Continental group (mostly located north of Madagascar and around Reunion island). This classification also resulted in a mixed zone where all three types of air mass are present in the middle of the Subtropical Indian Ocean.

## 5 Spatial and temporal variability of marine aerosols properties

### 5.1 Aerosol number concentration

Three modes were determined according to the average number size distribution of the filtered database derived from SMPS and OPC-N3 measurements: an Aitken and an Accumulation mode composed of sub-micron particles from 30 nm to 70 nm and from 71 nm to 350 nm (dry diameters) respectively, and a coarse mode made of super-micron particles of sizes ranging from 1.15 µm to 3.5 µm. Figure 4 shows the spatial variation of total number concentration Na (a), of the Aitken mode (b), of the accumulation (c) and of the coarse mode (d) along the vessel track during the six campaigns.





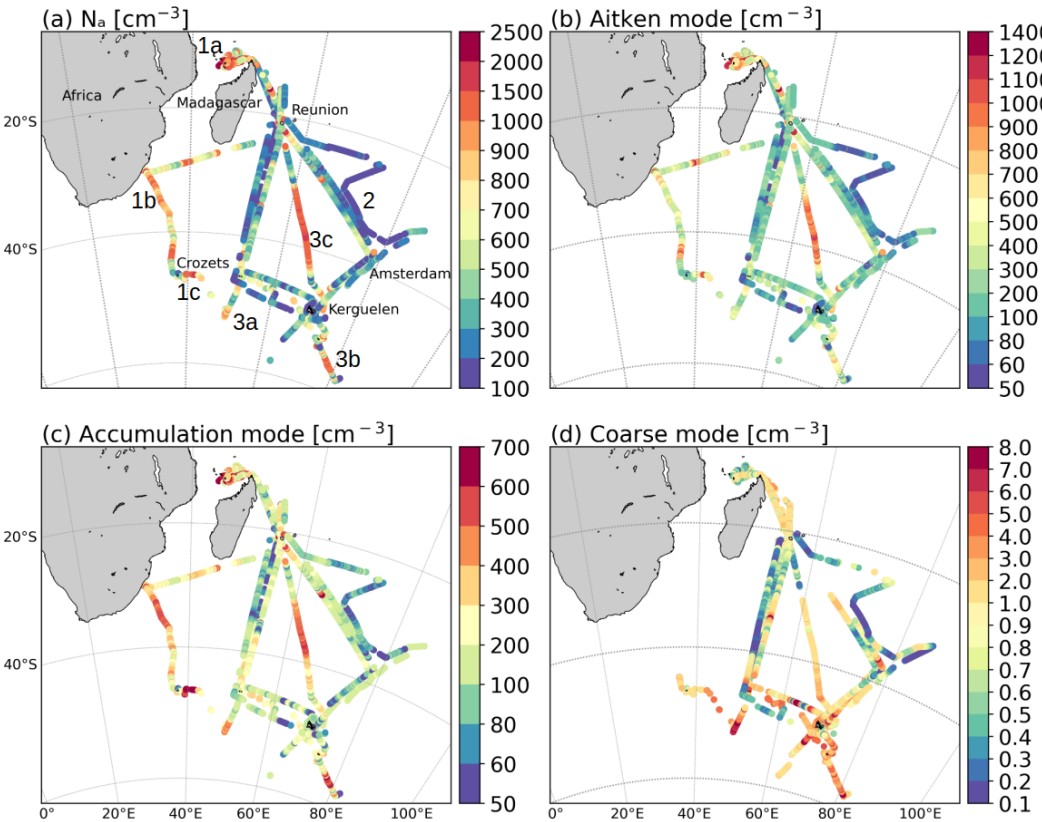

**Figure 4.** (a) Evolution of the total number concentration of aerosols measured by the SMPS from 30 nm to 350 nm, (b) in the Aitken mode (30 nm – 70 nm), (c) in the accumulation mode (71 nm – 350 nm), (d) and by the OPC-N3 in the coarse mode (1.15 μm – 3.5 μm) along the path of the Marion Dufresne in 2021 and 2023.

Na is highly variable on the sampling area with one order of magnitude between low and high concentrations ( 98 % of the values between $100 \, \mathrm{cm^{-3}}$ and $2500 \, \mathrm{cm^{-3}}$). In the Aitken mode, 97 % of the aerosol concentrations are between $50\text{-}1400 \, \mathrm{cm^{-3}}$

among which 10 % are smaller than $100 \, \mathrm{cm^{-3}}$, and  14 % are greater than $800 \, \mathrm{cm^{-3}}$. In the accumulation mode (Fig. 4 (c)), 93 % of the aerosol concentrations are between $50\text{-}700 \, \mathrm{cm^{-3}}$ among which 17 % are between $50 \, \mathrm{cm^{-3}}$ and $100 \, \mathrm{cm^{-3}}$ and  15 % are greater than $400 \, \mathrm{cm^{-3}}$. In the coarse mode (Fig. 4 (d)),  97 % of the aerosol concentrations are between $0.1\text{-} \, 8 \, \mathrm{cm^{-3}}$ within those less than  27 % are smaller than $0.5 \, \mathrm{cm^{-3}}$ and 10 % are greater than $3 \, \mathrm{cm^{-3}}$. Sedimentation and below-cloud scavenging are efficient removal processes of large particles of diameters greater than 500 nm (Saltzman, 2009; Bae et al., 2012). Plus,

the number of aerosols of diameters greater than 500 nm produced by bubble bursting is very low at source (Monahan et al., 1986; Ovadnevaite et al., 2014; Sellegri et al., 2023). This explains why the aerosol number concentration is lower in this mode and decreases during transport. Thus, the highest aerosol number concentrations in this mode are observed in regions where primary production is more important, as it is the case south of 40 ° S during the three storms identified in Figure 2a. The difference in the location of high and low aerosol concentrations reflects distinct underlying processes. Generally, an elevated



concentration of submicron particles above oceans can be explained by (i) a gas-particle conversion process implicating the homogeneous nucleation of sulfuric acid (Kulmala et al., 1998), (ii) by primary emission from bubble bursting (Clarke et al., 2003), (iii) by an entrainment of nanoparticles from the free troposphere (Zheng et al., 2021), or (iv) by advection of polluted air mass from the continent (Wang et al., 2020).

Na greater than 1000 $cm^{-3}$ are mostly observed north of Madagascar (Fig. 4 (a), label 1a). In this area [0° S-20° S; 40°
E-50° E] only 7 % of the data have Na below 500 $cm^{-3}$. The Aitken mode largely contributes to these high Na values and accounts on average for 61 % of Na. High concentrations of CO, CO2, CH4, and O3 were also observed (Tulet et al., 2024), suggesting a continental influence in these areas. These results are in-line with polluted air masses observed over the Atlantic Ocean during the Tara Pacific Expedition (Flores et al., 2020). Koponen et al. (2002) also reported a number concentration of aerosols greater than 2000 $cm^{-3}$ related to air masses coming directly from Europe during a cruise between the English
Channel and the coast of Antarctica.

Na concentration between 800-1000 $cm^{-3}$ are observed off the southeast coast of Africa (Fig. 4 (a), label 1b), but are not related to continental air masses as suggested by the 5-day back trajectories calculation, and are rather related to air masses coming from south of 40° S. Along this transect, the percentage of aerosols in the free troposphere is higher than the one in the marine boundary layer 2 days before the air masses arrive at the ship's location. This suggests an intrusion of free-tropospheric
air masses into the marine boundary layer.

Peaks of Na concentration are also visible east of Prince Edward islands (Fig. 4 (a), label 1c). Available measurements of Aerosol Optical Depth (AOD) over the period is between 0.141 and 0.893 (https://mobile.photons.univ-lille.fr, last access: 19 November 2024). According to Mallet et al. (2018), AOD values greater than 0.1 in the Indian Ocean indicate that the air mass has a continental origin and comes from South Africa or Australia. This hypothesis is likely as the air mass is located above
the marine boundary layer the day before it reaches the ship's location.

Na smaller than 200 $cm^{-3}$ (Fig. 4 (a), label 2) are mostly observed in the eastern part of the Indian Ocean [20° S-40° S; 60° E-80° E] during OBSAUSTRAL. This region is associated with weak to moderate wind speed (< 10 m s$^{-1}$) (Fig. 2a) and is far from any continental influence as the 5-day back trajectory air masses came from the Southern Indian Ocean (example in Fig. 11). In this region, 90 % of Na is smaller than 300 $cm^{-3}$ and exhibit low variations. Aerosol number concentration in the
Aitken and accumulation modes is mostly lower than 160 $cm^{-3}$ and shows weak variations in the Aitken mode. The coarse mode is associated with weak concentrations lower than 1 $cm^{-3}$ with most values between 0.1 and 0.6 $cm^{-3}$. These features suggest that the air masses arriving at the vessel may have aged, and that there were no significant sources of new particles. Another possibility is that the air mass may have experienced low wind speed or wash out.

During the three storms of SWINGS (Fig. 2 (a), black circles), Na is between 700-1000 $cm^{-3}$ (Fig. 4 (a), label 3a, 3b and
3c). The transects 3a and 3b are associated with the highest concentrations of aerosols in the coarse mode (5.56 ± 4.48 $cm^{-3}$ and 4.58 ± 3.70 $cm^{-3}$) compared with the transect 3b (1.85 ± 2.65 $cm^{-3}$). Plus, over the period of high wind speed (> 10 m s$^{-1}$), the average ratio of the Aitken mode concentration to the accumulation mode concentration is the highest along the transect 3b (1.93 ± 0.88), compared with the values along the transects 3a and 3b (1.12 ± 0.37 and 1.1 ± 0.31). In their study, Koponen et al. (2002) catched several episodes of high number concentration of aerosols (close or above 1000 $cm^{-3}$) between




Cape Town, South Africa, and 55° S, which were due to elevated concentrations in the Aitken mode, whereas between 55-60 ° S the high number concentration of aerosols were due to elevated concentrations in the nucleation mode. More recently, McCoy et al. (2021) described an aerosol mechanism occurring in the Southern Ocean which explains the high concentrations of Aitken mode particles in the marine boundary layer. This mechanism is triggered by emissions of dimethyl sulfide (DMS), which is uplifted by synoptic scale motions into the free troposphere. There, new particles can be formed by nucleation of DMS

oxidation products (mainly sulfuric acid), which constitute a reservoir of small aerosols above clouds. When free tropospheric air masses subside into the marine boundary layer, these small aerosols grow into CCN sizes via gas condensation and cloud processing and impact cloud droplet number.

## 5.2    Cloud condensation nuclei concentration

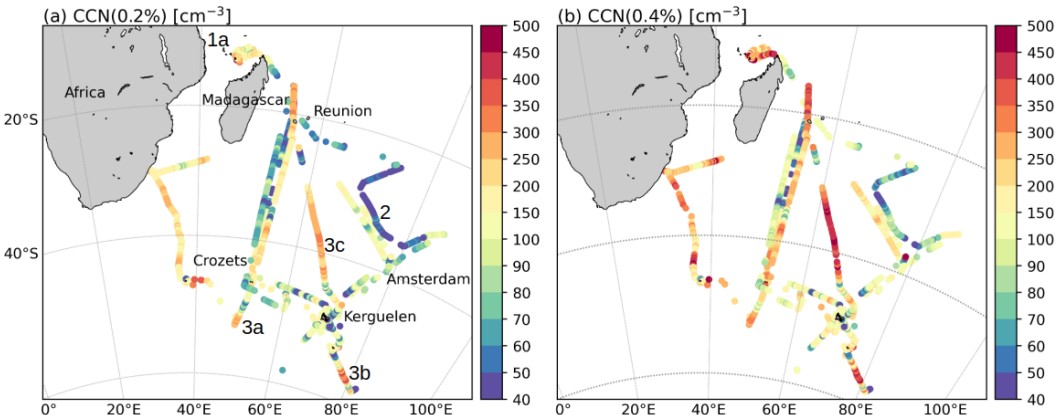

**Figure 5.** Evolution of the number of cloud condensation nuclei at 0.2 % (a) and 0.4 % (b) supersaturation along the path of the Marion Dufresne.

When it comes to discussing the activation properties of aerosols we use the term Condensation Nuclei (CN). CN is equivalent
to Na used previously and represents the total aerosols number concentration. Cloud Condensation Nuclei (CCN) refers to the subset of CN on which the atmospheric water vapor condenses upon at a given supersaturation (SS) to form cloud droplets. Thus CCN/CN gives the proportion of potential cloud droplets among the aerosol population. The variation of this ratio can be explained by a variation in the aerosol chemical composition and size distribution.

Fig. 5 shows the spatial variation of CCN concentration at 0.2 % and 0.4 % SS. 95 % of CCN number concentration is
between 40 and 500 cm$^{-3}$ at both SS. The average CCN concentration is 161.1 ± 82 and 190.5 ± 102.5 (mean ± one standard deviation) at 0.2 % SS and 0.4 % SS. These results are in the range of CCN concentrations reported by Sanchez et al. (2021) between Tasmania and Antarctica (45° S-70° S), where CCN concentrations are between  50-150 cm$^{-3}$ at 0.2 % SS, and between  90-230 cm$^{-3}$ at 0.4 % SS, regardless of the cloud processes affecting the air masses they analyzed.

Region 1a is associated with an average CCN concentration of 261.9 ± 188.4 cm$^{-3}$ and 426.3 ± 255.5 cm$^{-3}$ at 0.2 %
and 0.4 % SS, respectively. Along the transect 2, CCN concentration is mostly lower than 50 cm$^{-3}$ (100 cm$^{-3}$) at 0.2 % SS





(0.4 % SS). Such a low concentration in this area is not surprising since the aerosol number concentration in the accumulation mode, which contributes significantly to the CCN concentration, is already low (Fig. 4 (c)). In their study, Sanchez et al. (2021) attributed low Na and CCN concentration events to scavenging. Their measurements exhibit CCN concentrations lower than 100 $\mathrm{cm}^{-3}$ at 0.2 % and 0.4 % SS, and the lower CCN concentration at 0.3 % SS corresponded to the highest total precipitation.

Between 40-45 ° S and 45-50 ° S, the latitudinal distribution of CCN at 0.2 % SS (189.4 $\mathrm{cm}^{-3}$, 128.5 $\mathrm{cm}^{-3}$) is similar to the one obtained by Humphries et al. (2021) (169 $\mathrm{cm}^{-3}$, 131 $\mathrm{cm}^{-3}$). Between 50-55 ° S and 55-60 ° S, the CCN concentrations (169.2 $\mathrm{cm}^{-3}$, 185.1 $\mathrm{cm}^{-3}$) are higher than the one obtained by Humphries et al. (2021) (102 $\mathrm{cm}^{-3}$ in both latitudes range). This difference in the high latitudes range is certainly due to the low number of data collected in this area during MAP-IO.

Along the transects 3 (Fig. 5; label 3a, 3b, 3c), CCN concentration at 0.2 % SS is relatively similar. However, at 0.4 % SS,
CCN concentration is higher for transect 3b (299 ± 131 $\mathrm{cm}^{-3}$) and 3c (305 ± 134 $\mathrm{cm}^{-3}$), compared with the one for transect 3a (223 ± 94 $\mathrm{cm}^{-3}$). These transects are also marked by the development of an accumulation mode following the maximum wind speed, which allows CCN concentration to increase.





## 5.3 Activation diameters and hygroscopicity

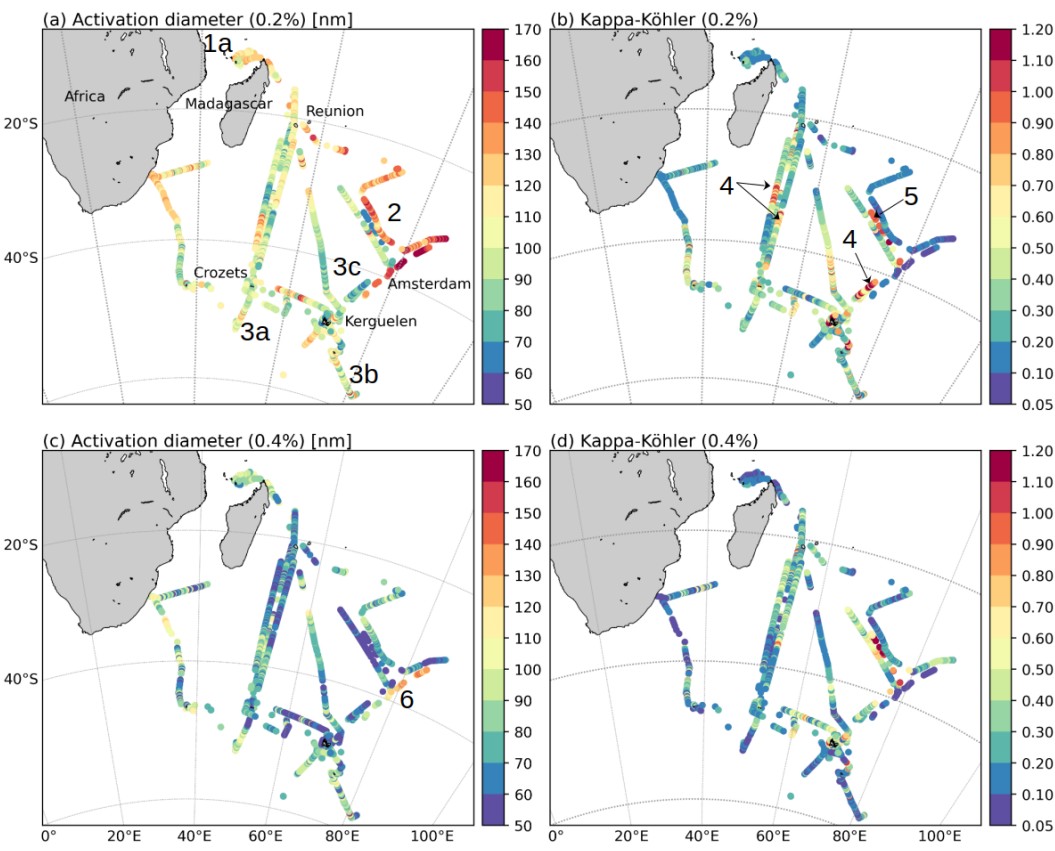

**Figure 6.** Evolution of the activation diameters ((a), (b)) and kappa parameter ((c), (d)) at 0.2 % and 0.4 % supersaturation along the path of the Marion Dufresne.

To further investigate the hygroscopic properties of the aerosol population, the spatial variation of the activation diameter and

$\kappa$ have been computed according to Petters and Kreidenweis (2007) (section 3.2). Fig. 6 presents the activation diameters (Fig. 6a and c)) and the corresponding $\kappa$ (Fig. 6b and d) at 0.2 (top panel) and 0.4 % SS (bottom panel). Activation diameters and $\kappa$ are highly variable along the different routes taken by the Marion Dufresne indicating a large variability of the chemical composition and size distribution of aerosols in the Southern Indian Ocean and the Southern Ocean. Globally lower $\kappa$ are observed when SS increases. 97 % of the $\kappa$ values are in the physical range between 0.05-1.2 at both SS, with an average value

of $0.36 \pm 0.2$ at 0.2 % SS (Fig. 6 (b)), and $0.25 \pm 0.18$ at 0.4 % SS (Fig. 6 (d)). The percentage of $\kappa$ values greater than 1 are negligible (1.5 % at 0.2 % SS and 0.9 % at 0.4 % SS). The average activation diameter is $104.1 \pm 19.7$ nm at 0.2 % SS (Fig. 6a), and $76 \pm 16.3$ nm at 0.4 % SS (Fig. 6c).



Higher activation diameters are observed north of Madagascar at both SS (Fig. 6 (a); label 1a). The associated $\kappa$ falls down to values between 0.05-0.2 at 0.4 % SS, corresponding to hydrophobic aerosols. This result corresponds to the most polluted air mass due to the proximity of continental emissions as mentioned before.

High activation diameters ( > 110 nm) are also observed along the transect 2 (Fig. 6(a)) at 0.2 % SS, which is in line with the previous observations of Na and CCN (Section 4.1 and 4.2) in this region.

During the storm events (Fig. 6; label 3a, 3b, 3c) (Fig. 1), a significant drop in $\kappa$ values is observed between 0.2 % SS and 0.4 % SS associated with primary emission. The average $\kappa$ calculated over a one-day storm is 0.35, 0.41, and 0.53 (Fig. 6 (b); label 3a, 3b, 3c) at 0.2 % SS. At 0.4 % SS, this value is 0.2, 0.19, and 0.23. The average $\kappa$ at 0.4 % SS (Fig. 6 (d)) falls in the range of $\kappa$ values of organic matter (0.1-0.2) according to Petters and Kreidenweis (2007). Hence, here the presence of organic matter could explain this decrease in the hygroscopicity, with a more visible effect on smaller aerosols (O'Dowd et al., 2004).

On the other hand, more hygroscopic species are observed along the transects between Reunion island and the Crozet islands (Fig. 6(b); label 4) during OP4, OBSAUSTRAL, and between Kerguelen islands and Amsterdam island during OP4 (Fig. 1). The average $\kappa$ is $0.8 \pm 0.3$ at 0.2 % SS and $0.6 \pm 0.3$ at 0.4 % SS. The activation diameter is already low at 0.2 % (70-90 nm) and $\kappa$ values are relatively high, mostly between 0.6 and 1. Here, the aerosol population seems to consist of both sulfates and organic species generated either by the local environmental conditions or transported from farther sources. Hence, over these periods, the wind speed is globally between 10-20 m s$^{-1}$, and the ship is influenced by air masses located above the marine boundary layer before arriving at its location.

Along the transect 5, high $\kappa$ values are observed at both SS, while the activation diameter decreases from 70 to 40 nm. During this period, the wind speed is between 5-17 m s$^{-1}$, the wave height is between 4-8 m, and the number concentration of aerosols in the coarse mode is between 0.7-1 cm$^{-3}$. Plus, the air mass stayed in the marine boundary layer. Here, the local weather conditions led to the emission of primary marine aerosols constituted of inorganic species.

Between Kerguelen islands and Amsterdam island during OBSAUSTRAL (Fig. 6 (c); label 6), low $\kappa$ values (0.1-0.2) are observed at both SS, and do not vary much from 0.2 % to 0.4 % SS. The air mass originated from Antarctica. Along this transect, Na decreases from 500 cm$^{-3}$ to 250 cm$^{-3}$ (Fig. 4a) and on the SMPS scan, the accumulation mode moves towards larger diameters with its lower diameter increasing from 100 nm to 200 nm. Thus this transect is a typical case of aerosol aging over time. The decrease in sea spray aerosols hygroscopicity due to aging has been widely studied and is discussed in Su et al. (2022). The authors focused on chloride depletion through air mass transport, generally observed when air masses originated from the continent. The magnitude of the chloride depletion increases following the order land-coastal area-remote ocean. Heterogeneous reactions between sea spray aerosols and atmospheric oxidants (e.g., O3, OH) and acidic species (e.g., nitrogen and sulfur organic species, organic acids) have been identified as playing a key role in chloride depletion during atmospheric transport.

At 0.2 % SS, the median (0.38) and interquartile values (0.26 and 0.52 for 25th and 75th quartiles) of $\kappa$ between 40-60° S are in good agreement with the results found in Tatzelt et al. (2022), in which the median value is 0.4, and the 25th and 75th quartiles are 0.3 and 0.65. Between 45-60° S, the 5° latitude bin averaged $\kappa$ values at 0.2 % SS (0.4 % SS) range



between 0.38-0.5 (0.1-0.2). In comparison, in Sanchez et al. (2021), these values globally range between 0.1-0.5 at 0.3 % SS, independently of the atmospheric processes affecting marine aerosols.

## 6  Aerosols size distribution based on their air mass origin

### 410  6.1  Size distribution of marine aerosols

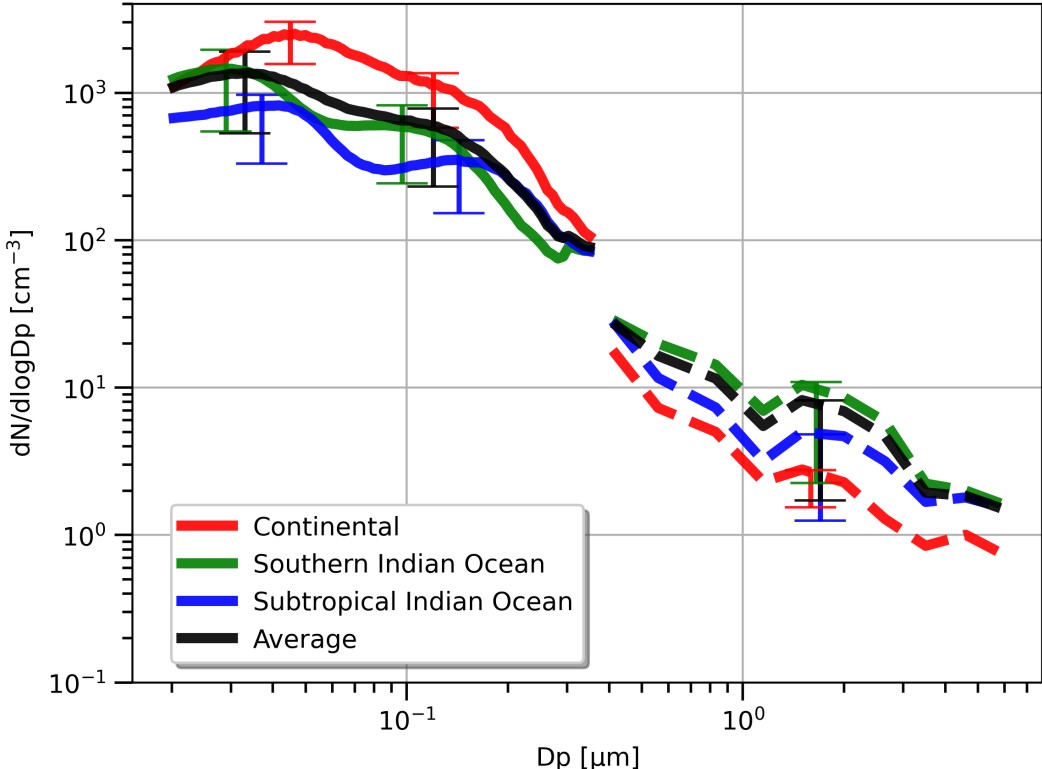

**Figure 7.** Average size distributions of aerosols according to air mass origin (continental, Southern Indian Ocean or Subtropical Indian Ocean) of the 5-days back-trajectories simulated by the FLEXPART model. Solid lines are average size distributions derived from SMPS data and dashed lines are average size distributions derived from OPC-N3 data. Bottom and top limits of error bars centered in the mean geometric diameter of each mode are the 25th and the 75th quartiles.



| | Continental | Southern Indian Ocean | Subtropical Indian Ocean | Average |
|---|---|---|---|---|
| Aitken mode | D = 45 nm | D = 29 nm | D = 37 nm | D = 33 nm |
| | $\sigma$ = 1.52 | $\sigma$ = 1.5 | $\sigma$ = 1.54 | $\sigma$ = 1.73 |
| | N= 1100 cm$^{-3}$ | N = 650 cm$^{-3}$ | N = 400 cm$^{-3}$ | N = 830 cm$^{-3}$ |
| | RMSE = 19.2 cm$^{-3}$ | RMSE = 13.9 cm$^{-3}$ | RMSE = 10.2 cm$^{-3}$ | RMSE = 15.1 cm$^{-3}$ |
| Accumulation mode | D = 120 nm | D = 97 nm | D = 143 nm | D = 120 nm |
| | $\sigma$ = 1.56 | $\sigma$ = 1.56 | $\sigma$ = 1.52 | $\sigma$ = 1.52 |
| | N= 490 cm$^{-3}$ | N = 285 cm$^{-3}$ | N = 160 cm$^{-3}$ | N = 240 cm$^{-3}$ |
| | RMSE = 16.9 cm$^{-3}$ | RMSE = 10.5 cm$^{-3}$ | RMSE = 6.7 cm$^{-3}$ | RMSE = 12.4 cm$^{-3}$ |
| Coarse mode | D = 1.59 μm | D = 1.65 μm | D = 1.7 nm | D = 1.7 μm |
| | $\sigma$ = 1.53 | $\sigma$ = 1.56 | $\sigma$ = 1.5 | $\sigma$ = 1.5 |
| | N= 1.17 cm$^{-3}$ | N = 4.5 cm$^{-3}$ | N = 2.2 cm$^{-3}$ | N = 3.3 cm$^{-3}$ |
| | RMSE = 0.5 cm$^{-3}$ | RMSE = 1.05 cm$^{-3}$ | RMSE = 0.69 cm$^{-3}$ | RMSE = 0.93 cm$^{-3}$ |

**Table 2.** Geometric parameters of each mode of aerosol average size distributions (mean diameter, log-normal standard deviation and total number concentration).

Figure 7 shows the average aerosol size distributions as a function of air mass origin, calculated by FLEXPART using the method described in section 3.3. Table 2 presents the corresponding geometric parameters of each mode fitted on a log-normal distribution. For the SMPS data (0.02-0.35 μm) 6273, 18647, and 9049 data were used in the calculation of the average aerosol size distribution of the continental, Southern Indian Ocean, and Subtropical Indian Ocean air masses. For the OPC-N3 data

(0.41-5.85 μm), 2305, 13727, and 5554 data were used in the calculation of the average size distribution of the continental, Southern Indian Ocean, and Subtropical Indian Ocean group. Independently of the origin of air masses, three modes are visible in the size distribution. The average mean diameters of the Aitken, accumulation, and coarse modes are 33 nm, 120 nm, and 1.7 μm and the standard deviations are 1.73, 1.52, and 1.5. Two other modes were also detected between 300 nm and 1 μm but are not considered in the discussion as their position is in the upper limit detection of the SMPS and the lower limit detection of the

OPC-N3 respectively. The differences in the coarse mode are more visible in the number concentration of aerosols rather than the mean diameters and standard deviation. The continental, Subtropical Indian Ocean, and Southern Indian Ocean air masses exhibit a number concentration of aerosols of 1.17, 4.5, and 2.2 cm$^{-3}$ (average number concentration of 3.3 cm$^{-3}$) with a mean diameter of 1.59, 1.65, and 1.7 μm (average position at 1.7 μm). The number concentration of aerosols is the greatest for the Southern Indian Ocean air masses and the lowest for the continental air masses, consistent with the presence of regular

strong winds and swell in the southern Indian Ocean that generates significant primary emission. The average wind speeds calculated are 11.2, 8.8, and 4.3 m s$^{-1}$ for the Southern Indian Ocean, Subtropical Indian Ocean, and continental group and 86 % of strong wind speed (> 20 m s$^{-1}$) events are found in the Southern Indian Ocean group. The differences in the sub-micron



modes between different air mass types are visible both in the number concentration of aerosols, the mean diameters, and the standard deviation of the modes. Average number concentration of aerosols is 830 $\text{cm}^{-3}$ in the Aitken mode and 240 $\text{cm}^{-3}$

in the accumulation mode. Continental air masses exhibit the greatest number concentration of aerosols in the Aitken and accumulation modes with 1100 and 490 $\text{cm}^{-3}$, respectively. Subtropical Indian Ocean air masses exhibit the lowest number concentration of aerosols in the Aitken and accumulation modes with 400 and 160 $\text{cm}^{-3}$. Larger mean diameters are observed in the continental and Subtropical Indian Ocean groups with a similar value for the Aitken mode (45 and 37 nm) and values of 120 and 143 nm for the accumulation mode. Southern Indian Ocean air masses are characterized by lower mean diameters of

29 and 97 nm for the Aitken and accumulation modes. Considering that both Subtropical Indian Ocean and Southern Indian Ocean air masses are marine and minimally affected by terrestrial aerosols, the shift in the mean diameters of Subtropical Indian Ocean towards larger sizes may indicate that aerosols constituting this group have aged: the processes of coagulation and condensation had time to grow aerosols to larger sizes. Plus, the larger N in the Aitken mode of the continental air masses implies that there will be a more important coagulation process and therefore a less marked discontinuity between the Aitken

and accumulation modes compared with cleaner air masses of the Subtropical Indian Ocean group. On the opposite end, the position of the sub-micron mode mean diameters of the Southern Indian Ocean group indicate newer primary marine aerosol emissions and less aging. The position of the sub-micron modes are in good agreement with the results obtained by Xu et al. (2021) and Kawana et al. (2022) who compared marine aerosol size distributions under continental influences and far from continental sources. In Kawana et al. (2022), the size distribution of terrestrial air masses exhibits a weak demarcation between

the Aitken and the accumulation mode making the size distribution more monomodal. For marine air masses, a clear bimodal size distribution is visible with a lower number concentration of aerosols in the Aitken and accumulation mode compared with terrestrial air masses. Xu et al. (2021) showed that clean air masses with high and low levels of biological activity exhibited a bimodal size distribution with an Aitken mode mean diameter ranging between 30-50 nm and an accumulation mode mean diameter between 100-200 nm. In their study, polluted air masses with high and low levels of biological activity showed a more

monomodal size distribution with a peak mean diameter between 50-60 nm.

## 6.2 Relationship between marine aerosol hygroscopicity, wind speed, and nanophytoplankton abundance

Sellegri et al. (2021) investigated the link between sea spray aerosol (SSA) fluxes and the abundance of phytoplankton cells of different nature (nanophytoplankton, picoplankton and coccolithophore-like cells) in surface seawater at different temperature. They found the strongest linear correlation between SSA and nanophytoplankton cells. In this section, the lowest and highest

5 % of $\kappa$ values are not considered for the sake of statistical representativeness.





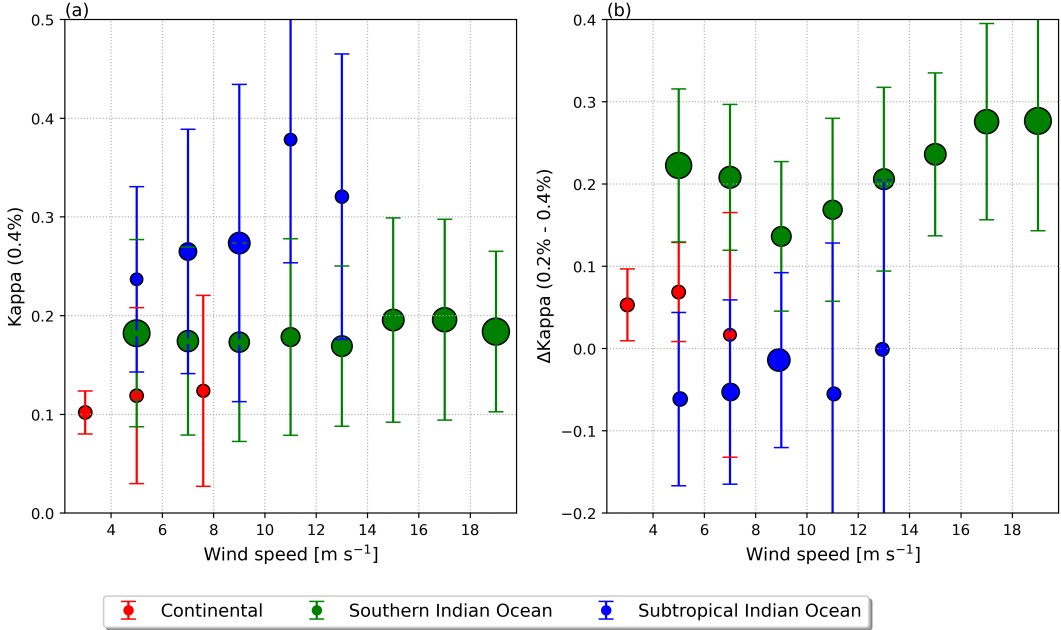

**Figure 8.** (left side) $\kappa$ at 0.4 % SS against wind speed. (right side) Difference between $\kappa$ at 0.2 % SS and 0.4 % SS against wind speed. Circles are the $\kappa$ values averaged every 2 m s$^{-1}$. Nanophytoplankton abundance is proportional to the size of the circles in cells.

Figure 8a shows the evolution of $\kappa$ values at 0.4 % SS as a function of local wind speed, local nanophytoplankton abundance and the three types of air mass. The average nanophytoplankton abundance values are more elevated for the Southern Indian Ocean group with an average value of $846.4 \pm 518.9$ cells mL$^{-1}$, compared with average values of $282.3 \pm 120.0$ cells mL$^{-1}$ for the Subtropical Indian Ocean group. On the left panel, the separation between the three types of air masses is clearly visible, with the highest average $\kappa$ values (0.24-0.4) for the Subtropical Indian Ocean air masses, the smallest values for the Continental air masses (0.1-0.13), and intermediate values for the Southern Indian Ocean air masses (0.17-0.22). The effect of wind speed on hygroscopicity is clearly visible for the Subtropical Indian Ocean, with $\kappa$ values increasing from 0.24 to 0.37 for wind speeds between 3 and 12 m s$^{-1}$. This result is consistent with the low nanophytoplankton abundance in this group, which makes the $\kappa$ values closer to NaCl $\kappa$. On the contrary, no trend between wind speed and $\kappa$ is observed for air masses of the southern Indian Ocean group. Figure 8b shows the difference in $\kappa$ between supersaturation at 0.2 and 0.4 % according to wind speed, phytoplankton abundance and air mass type. This difference in $\kappa$ indicates the evolution of hygroscopicity for small particles between the two activation diameters. It is noteworthy that the Southern Indian Ocean group stands out from the two others. Hence, the difference in $\kappa$ between 0.2 % and 0.4 % SS according to the wind speed is the lowest (in average around 0) for the Continental and Subtropical Indian Ocean air masses, whereas it is between 0.1 and 0.3 for the Southern Indian Ocean air masses, with an increase in this difference when the wind speed increases from 9 to 17 m s$^{-1}$. One difference between the three groups which could explain these features could be a significant amount of organic matter present in the aerosol of the Southern Indian Ocean group. Therefore the difference in $\kappa$ for small particles of diameter between 76 nm and



104 nm is more pronounced in the regions of active biological production. The increase in the organic fraction towards smaller aerosol sizes has already been highlighted in O'Dowd et al. (2004), and is either possible due to the presence of primary marine

organic aerosols (Schwier et al., 2015) or to the presence of volatile organic compounds which condenses in larger proportions on small sea salts. In the case of primary marine aerosols, O'Dowd et al. (2004) attribute the increasing organic enrichment of submicron aerosols with decreasing size to bubble-bursting processes when the ocean surface layer is more concentrated with surfactants which can be adsorbed at sea spray droplets surface. This is in agreement with previous studies (e.g. (Oppo et al., 1999)). In the case of secondary marine aerosols, the surface-to-volume ratio increases when the size of aerosols decreases and

provides the condensation of organic volatile compounds onto submicrometer particles (Mayer et al., 2020).

## 7    Focus on particular events

Among the numerous situations observed during the 6 campaigns presented in this paper, several deserve to be detailed in order to analyze their origins or specific processes. Three types of situations have been selected (Fig. 9) which led to high aerosol observations with peaks above $3000 \, \mathrm{cm}^{-3}$. The first corresponds to a polluted air mass observed during the SCRATCH

campaign in northern Madagascar (Section 6.1). The second situation corresponds to a nucleation case of new particles in the marine atmospheric boundary layer observed during the OP3 campaign (Section 6.2). Finally, the last situation selected corresponds to a storm situation observed during SWINGS (Section 6.3), highlighting the primary production of aerosols and the evolution of their size distribution with air mass aging.




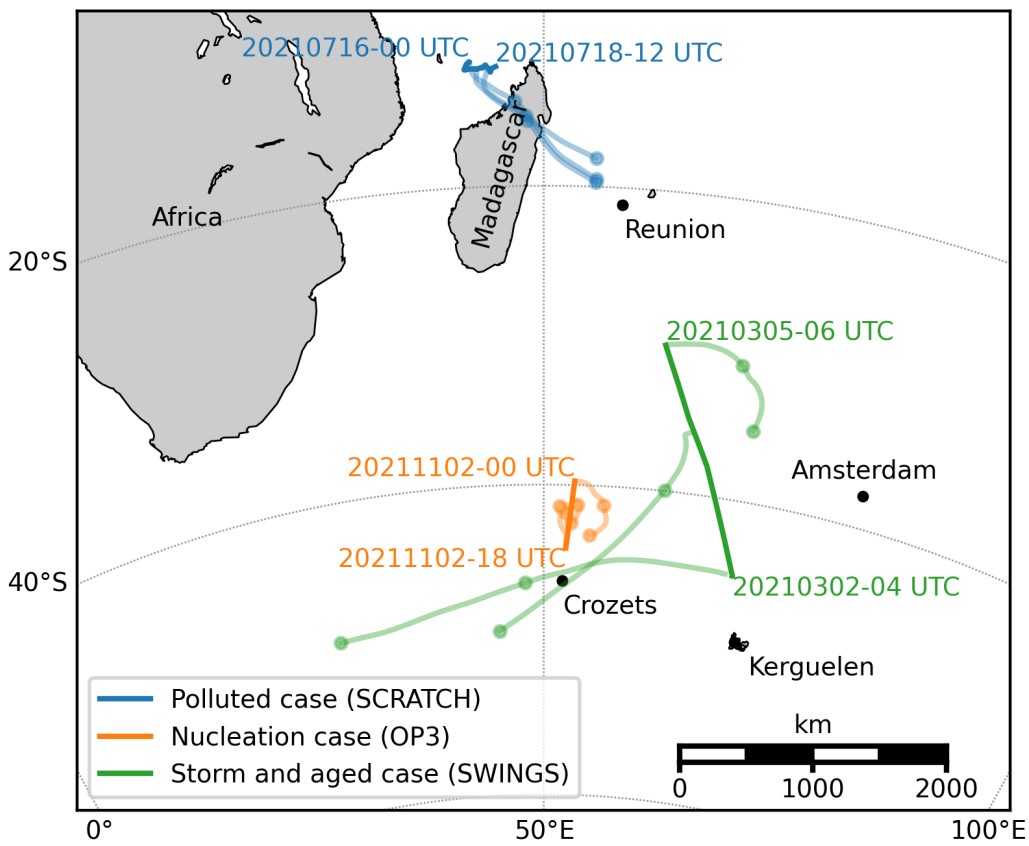

**Figure 9.** Position of the Marion Dufresne (thick lines) and 2-day back-trajectories (thin lines) from the FLEXPART model for the three selected situations, illustrating the origins of the air masses (1 dot per day).

## 7.1 Polluted case (SCRATCH)

Figure 10 shows the evolution of aerosol size distribution (e) and total aerosol and CO concentrations (f) between July 16 and 19, 2021. During this period, CO and aerosol concentration remains high (> 50 ppb and > 2000 $\mathrm{cm^{-3}}$, respectively), showing that the area is generally affected by residual continental pollution. Several Na peaks exceeding 5000 $\mathrm{cm^{-3}}$ are also measured nighttime and are associated with a significant increase in CO concentrations by 10 to 15 ppb, indicating that the air mass is affected by additional pollution. Aerosol size distribution shows that the high average concentration observed during the

period comes from the Aitken mode (30 to 40 nm). It can also be seen that each peak is associated with the transport of a new nanoparticles mode ( 5 to 30 nm) and the growth of the Aitken mode to reach the accumulation mode (30 to 100 nm). Back-trajectory analysis of Figure 9 (a,b,c,d) clearly indicates the passage of the air mass over the urbanized region of Majunga, located in north-western Madagascar. The altitude of the back-trajectory clearly indicates that the air mass sampled by the vessel is located above Madagascar between 800 m a.g.l. in the coastal zone and 1200 m a.g.l. in the center of the island, then



subsides over the sea during transport. The time evolution of the mixed boundary layer (MBL) thickness over Madagascar,
corrected for transport time, is shown in Figure 10g. There is an exceptional correlation of the MBL thickness with CO and Na
peaks: each concentration peaks are associated with a MBL thickness greater than 800 m. This means that the pollution emitted
over Madagascar has to be mixed in an atmospheric layer above 800 m to reach the ship. This can only happen in the afternoon
when the MBL is sufficiently developed. Note that during this period, there is no rain at the Marion Dufresne location or along
the back-trajectories to Madagascar (not shown here).

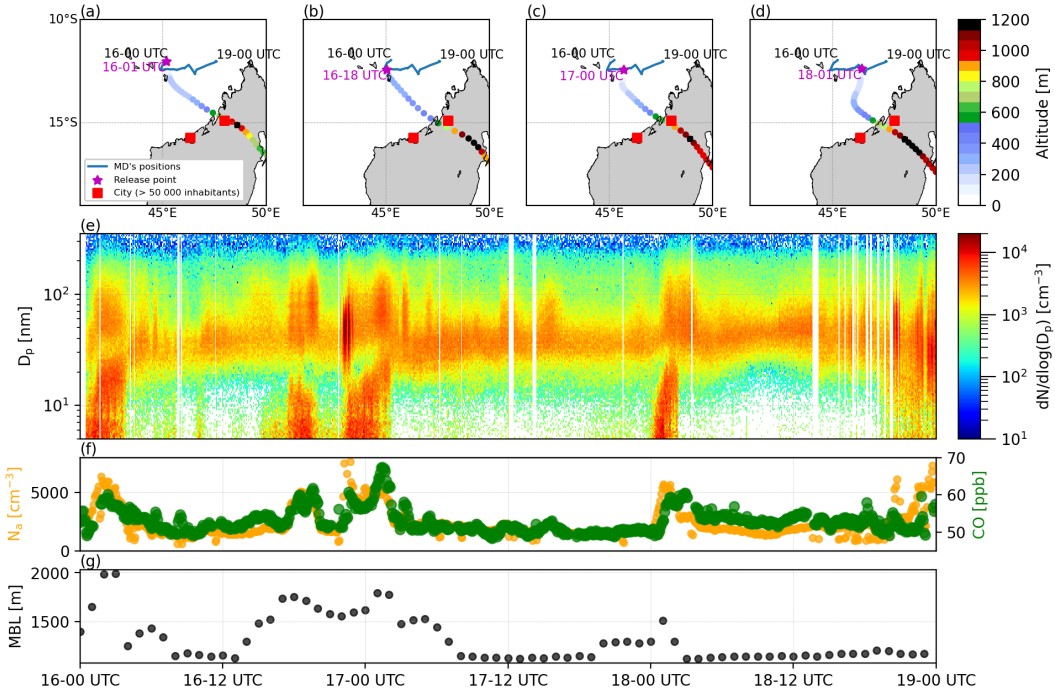

**Figure 10.** Temporal evolution of aerosol size distribution (e), aerosol number concentration (Na) and CO concentration (f) observed from
July 16 to 19, 2021 (SCRATCH campaign, Fig. 1). Mixed boundary layer thickness evolution during 3 first hours over West of Madagascar
along FLEXPART back-trajectories is presented in (g). Altitude of air masses along FLEXPART back-trajectories for the four CO peaks is
visible in (a : 16-00 UTC, b : 16-18 UTC, c : 17-00 UTC, d : 18-01 UTC). The red squares correspond to the locations of the most urbanized
areas in Majunga province (Madagascar).

## 7.2  Nucleation case (OP3)

Figure 11 shows the evolution of aerosol concentration (total number, size distribution and CCN) and ship track overlaid on
cloud mask based on satellite brightness temperature (cloud areas correspond to brightness temperature lower than 282 K)
(Janowiak et al., 2017; Wang et al., 2017) on September 11th, 2021.



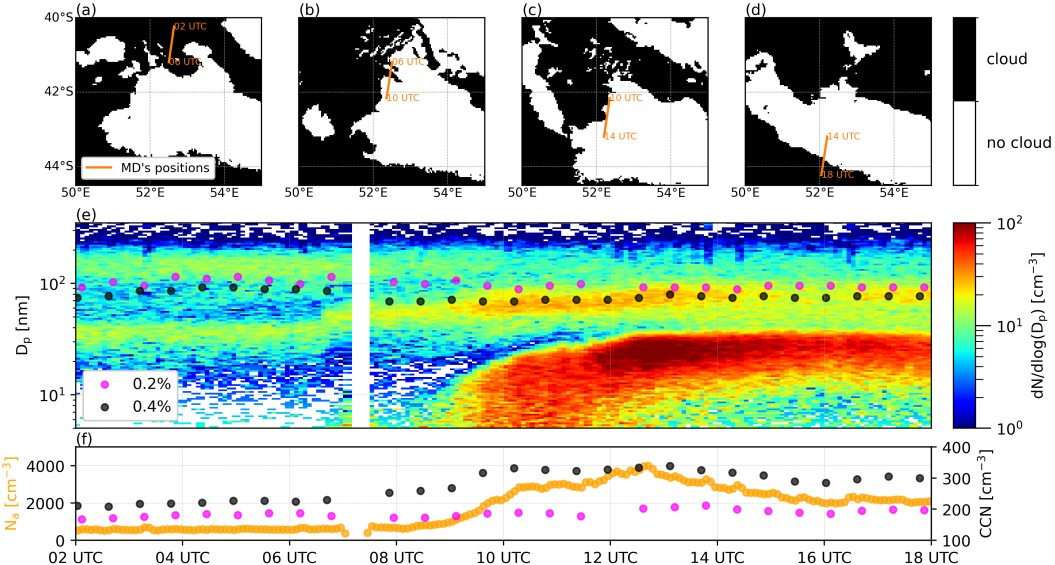

**Figure 11.** Cloud cover and Marion Dufresne positions in (a, 02-06 UTC), (b, 06-10 UTC), (c, 10-14 UTC) and (d, 14-18 UTC). Temporal evolution of (e) aerosol size distribution and activation diameter at 0.2 and 0.4 % supersaturation and (f) aerosol concentration and cloud condensation nuclei at 0.2 and 0.4 % supersaturation observed during November 2, 2021 (OP3 campaign, Fig. 1).

Before 08:00 UTC ( 12 local time (LT)), SMPS measurements clearly showed two modes at 30 nm and 130 nm associated with a total concentration of 500 $cm^{-3}$. The measured CCN number is low at 150 $cm^{-3}$ and 200 $cm^{-3}$ at 0.2 and 0.4 % of SS, respectively. At 09:00 UTC, a strong increase of the total aerosols is observed reaching 4000 $cm^{-3}$ at 13:00 UTC. The SMPS showed a formation of new particles at 09:00 UTC (5 nm) associated with a classical banana-shaped growth (condensation-coagulation) typical to nucleation event ((Kulmala and Kerminen, 2008; Foucart et al., 2018; Määttänen et al., 2018), and

references therein). The activation diameters at 0.2 and 0.4 % SS are larger than the size of the nucleation mode. On the other hand, in parallel with the nucleation process, we observe a growth of the Aitken mode which goes from 30 nm to 80 nm and exceeds the activation diameter at 0.4 % SS. Thus the number of CCN at 0.4 % SS will go from 220 $cm^{-3}$ (08:00 UTC) to 350 $cm^{-3}$ (10:00 UTC) while the number of CCN at 0.2 % SS will remain globally stable during the day. Both processes of new particle formation and growth result from the formation of secondary oxidation products (like the oxidation of DMS and

methanethiol (MeSH) in sulfuric acid) and are observed when UV is sufficient to form atmospheric oxidants (like hydroxyl radicals). From the satellite images in Figure 11 one can observe that the ship is located in a cloudy area until 07:00 UTC (11:00 LT) then passes into a clear sky area between 09:00 UTC and 15:00 UTC to pass again into a cloudy area. This passage corresponds exactly to the period of formation and growth of the particles. It is likely that the air mass was loaded with gaseous precursors before 07:00 UTC, which rapidly oxidized when the ship passed through clear skies, allowing the formation of

low-volatile species.



## 7.3 Storm and aged case (SWINGS)

The last period highlighted corresponds to a case of primary emission of aerosols retained during the SWINGS campaign between March 2 at 00:00 UTC and 6 March at 00:00 UTC 2021. In Figure 12a,b,c,d are presented the back-trajectories of the air mass passing over the Marion Dufresne on 2 March at 06:00 UTC, 3 March at 00:00 UTC, and 3 March at 18:00 UTC
projected onto a rain field (blue isolines) from the ERA5 reanalyses. The color of the back trajectory corresponds to the 10 m wind speed of the ERA5 fields along the trajectory. Figure 12d,e,f shows the temporal evolution of the size distribution, wave height, surface wind, total aerosol concentration, and hygroscopicity ($\kappa$).

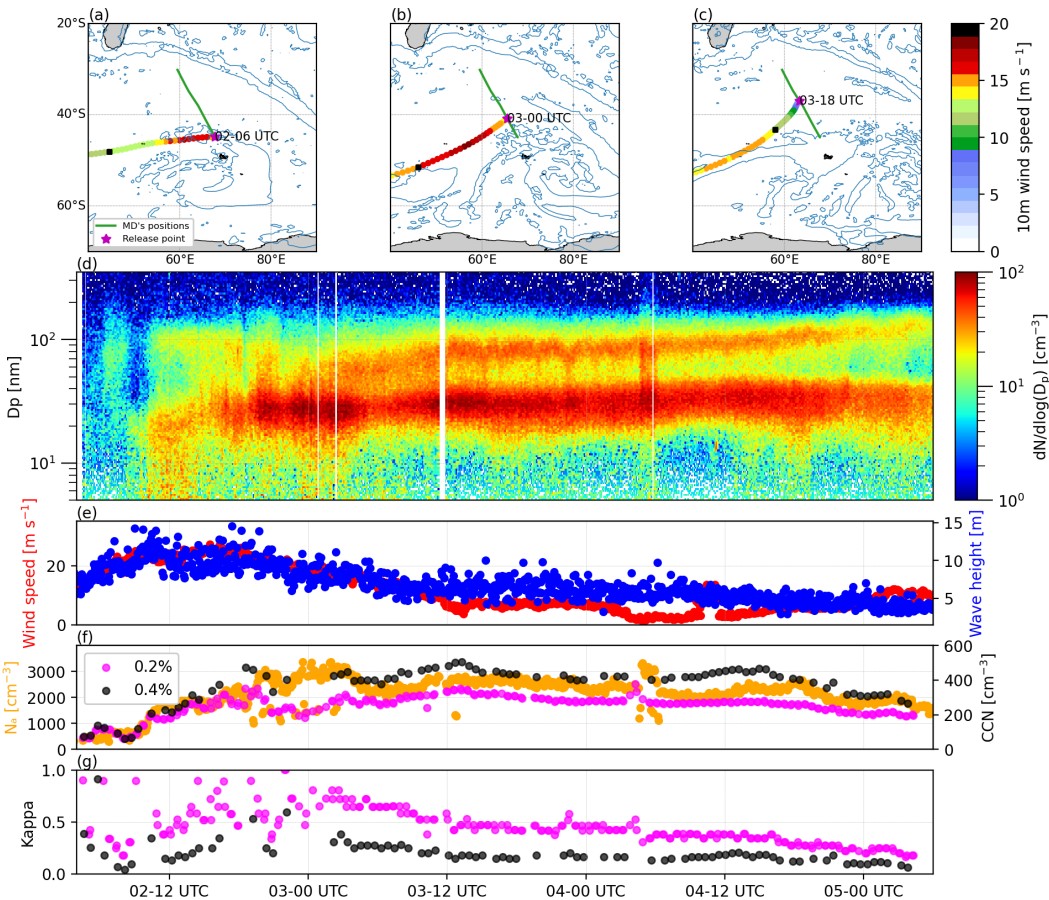

**Figure 12.** 10 m wind speed from ERA5 along 3 FLEXPART back-trajectories at three instant in (a), (b) and (c). Blues contours correspond to areas where rain rate is higher than 0.1 mm h$^{-1}$ at that instant. Temporal evolution of (d) aerosol size distribution, (e) wind speed and wave height, (f) aerosol concentration and cloud condensation nuclei at 0.2 and 0.4 % supersaturation and (g) hygroscopicity parameters at 0.2 and 0.4 % supersaturation observed from March 2 to 5 2021 during SWINGS campaign (Fig. 1).





Three periods are identifiable. On March 2 before 06:00 UTC, the air mass came from the west in a windstorm zone with ERA5 wind speeds greater than 18 m s$^{-1}$ over the previous 12 hours (Fig. 12a). The ship is affected by this storm since the

measured swell exceeds 14 m and the wind speed reaches 25 m s$^{-1}$. The air mass is however affected by rain which significantly limits aerosol concentrations particularly before 03:00 UTC ( 500 cm$^{-3}$). Hygroscopicity varies greatly with a measured $\kappa$ fluctuating between 0.3 and 1. The second period from 2 March at 18:00 UTC and 3 March at 06:00 UTC corresponds to a phase of decreasing wind and swell conditions (Fig. 12b,d). On 2 March at 18:00 UTC, the Marion Dufresne moved approximately 300 km towards the northwest. The measured wind and swell conditions remain very high and the air mass has remained in

heavy wind conditions greater than 20 m s$^{-1}$ over the last 24 hours. However, according to ERA5 analyses, this air mass has not been affected by rain. Consequently, the aerosol concentrations measured on the Marion Dufresne exceed 3000 cm$^{-3}$. Aerosol hygroscopicity has increased but remains quite fluctuating with a $\kappa$ varying between 0.5 and 1 (Fig. 12g). The third period begins on 3 March at 06:00 UTC. SMPS data show the maintenance of a pronounced Aitken mode at around 30 nm and a concentration of around 2000 cm$^{-3}$ (Fig. 12d,f,g). There is also the formation of a second mode whose size increases over

the period to reach a size characteristic of the accumulation mode at around 100 nm. During the period, the Marion Dufresne continued its movement towards the northwest away from the storm area. On 3 March at 18:00 UTC, ERA5 data show that the air mass was not affected by the rain and experienced moderate winds around 10 m s$^{-1}$ during the previous day (Fig. 12c). Observations on board the Marion Dufresne show that sea conditions remained rough at least until 4 March at 06:00 UTC with swell fluctuating between 6 and 10 m (Fig. 12e). While the total aerosol concentration remained broadly stable over this period,

it is notable that as the air mass aged, the hygroscopicity parameter gradually dropped from 0.7 on 3 March at 06:00 UTC to 0.2 on 5 March at 00:00 UTC (Fig. 12f,g).

## 8 Conclusions

Aerosol measurements collected during six oceanographic campaigns carried out in 2021 and 2023 within the Indian Ocean and the Southern Ocean on board the Marion Dufresne are presented and analyzed in this paper. SMPS data are used to study

the spatial distribution of Na over the sample area. Na shows a large variability with the highest concentrations observed either in tropical coastal regions or in the remote ocean and are attributable to large scale or local weather influences. CCN-100 measurements are used in collocation with SMPS measurements to study the activation ratio, activation diameter and hygroscopicity of aerosols at 0.2 and 0.4 % SS. The variability of the activation ratio allows to highlight differences in the chemical composition and size of the aerosols, which is confirmed by the spatial and temporal variability of the activation

diameters and $\kappa$ calculated according to Petters and Kreidenweis (2007). Globally the smaller aerosols activated between two activation diameters at 0.2 % and 0.4 % SS are more hydrophobic than those activated above the activation diameters at 0.2 % SS. $\kappa$ values of organic species (0.22-0.24) are observed at 0.4 % SS during storm events occurring south of 35° S. In these areas, the nanophytoplankton cell abundance is more important than north of 35° S. As local weather conditions cannot fully explain the calculated $\kappa$, we performed 5-day back trajectories to access the air mass's main origin: continental,

Subtropical Indian Ocean, or Southern Indian Ocean. The size distribution of aerosols according to the air masses origin



shows significant differences that have already been highlighted in previous studies. We found that the aerosols transported by Subtropical Indian ocean air masses are more aged than that of Southern Indian Ocean air masses, which is traduced by larger Aitken and accumulation mode mean geometric diameters. The largest (lowest) number concentration of aerosols in the Aitken and accumulation modes is observed for continental (Subtropical Indian Ocean) size distribution. The highest numerical

concentration of coarse mode aerosols is observed for the Southern Indian Ocean size distribution and is associated with rougher sea states and weather conditions (Tulet et al., 2024). Additionally, from this classification, we investigated the possible relationship between the wind speed, the aerosol hygroscopicity ($\kappa$), and the nanophytoplankton abundance. Continental air masses are associated with more hydrophobic aerosols ($\kappa$ from 0.1 to 0.13), whereas Subtropical Indian Ocean air masses are associated with more hydrophilic aerosols ($\kappa$ from 0.24 to 0.4). Southern Indian Ocean air masses exhibit in-between values

($\kappa$ from 0.17 to 0.22). $\kappa$ values of Subtropical Indian Ocean increase when the wind speed is getting stronger (from 5 to 12 m s$^{-1}$). A significant drop in $\kappa$ (between 0.2 % and 0.4 % supersaturation) when the wind speed increases from 9 to 17 m s$^{-1}$ is observed for Southern Indian Ocean air masses, and could be attributed to the presence of a more important concentration of organic species at the surface of the smaller aerosols. Finally, we focused on three situations. A polluted case during which high Na and CO concentrations are observed north of Madagascar during the night, and coincide with the passage of air masses

into a well developed boundary layer over Madagascar before their subsidence towards the ship's location. A nucleation case followed by the growth of the particles which occurred north of Crozets islands, and coincides with the time when the Marion Dufresne passed from a cloudy to a clear sky area. A period between Kerguelen islands and Reunion island during which the ship encountered storm conditions, and was under the influence of air masses affected by rain, leading to a decrease in the aerosol number concentration. As the ship moved toward Reunion island, it went away from the strong wind and wave height

conditions, and the air mass arriving at the ship's location were more aged, allowing the development of a stable Aitken mode, and an accumulation mode whose mean diameter moves towards larger diameters. These results give a view of the diversity of marine aerosols present in the Indian and the Southern Ocean and highlight their hygroscopicity and CCN properties. This diversity could be captured thanks to measurements realized over a long period of time and in various environmental conditions, which supports the interest of the MAP-IO program. The present work is based on 1.5 years of aerosol measurements and could

further be developed in the future with a larger database.




## Appendix A: Air masses classification

### A1

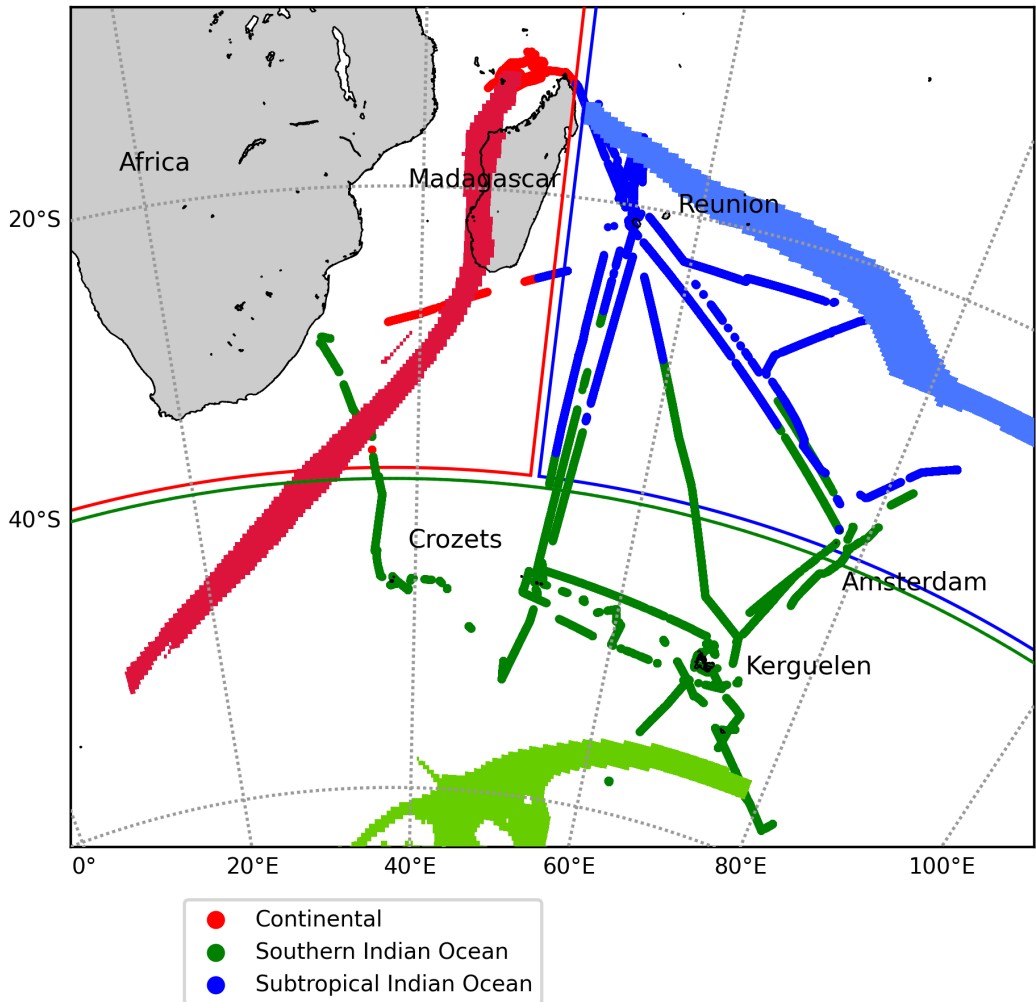

**Figure A1.** Classification of the aerosol data recorded in 2021 and 2023 according to the 5-days back-trajectories simulated by the FLEX-PART model. Back-trajectories in red, blue and green are examples of continental, Subtropical Indian Ocean and Southern Indian Ocean air masses.

*Data availability.* Atmospheric data are available on the AERIS datacenter: https://www.aeris-data.fr/ (last access: 1 December 2024). Cytometry data are available on the SEANOE datacenter: https://www.seanoe.org/data/00783/89505/ (last access: 1 December 2024)



*Author contributions.* PT is the head of the MAP-IO program. JMM have been in charge of the installation and the maintenance of the instruments on-board. PT, JB are responsible for the aerosols in-situ data. MT supervised the installation of the Cytosense onboard the $Marion Dufresne$. MT is responsible for the Cytosense scientific operations and instrument maintenance. MT analyzed and provided the phytoplankton data set. MD, PT are responsible for the aerosol data treatment. JP, JB set up the FLEXPART back-trajectories simulations. MD, PT, JP worked on the analysis of the aerosol, weather and phytoplankton in-situ data, and the FLEXPART outputs. MD, PT, JP worked
on the paper's figures.

*Competing interests.* The contact author has declared that none of the authors has any competing interests.

*Acknowledgements.* The authors highly acknowledge the TAAF, IFREMER, LDAS, and GENAVIR for their help and constant support in the installation and the maintenance of all scientific instruments on board the Marion Dufresne. The authors also thank the technical teams of the LACy and OSU-R engaged in the data acquisition and the maintenance of the instruments of the MAP-IO program and the financial
and human support of each laboratory partners such as OSU-R, LACy, LaMP, LAERO, LOA, LATMOS, LSCE, MIO, and ENTROPIE. MAP-IO is a scientific program led by the University of La Réunion and was funded by the European Union through the ERDF programme, the University of La Réunion, the préfecture de la Réunion, the Région Réunion, the CNRS, IFREMER, and the Flotte Océanographique Française. The authors also acknowledge Cyrielle Denjean and Sophia Brumer for the scientific discussions, and Vincent Noël for providing the satellite data.



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
