# Peer review of "Origin, size distribution and hygroscopic properties of marine aerosols in the south-western Indian Ocean: report of 6 campaigns of shipborne observations"

_EGUsphere, 2024_

## Author Response (AR1)

**Referee 1:**

General comments:

This manuscript by Dournaux et al. provides the characteristics on aerosol number concentrations, particle size distributions, CCN number concentrations and their hygroscopicity in several ocean regions, and discusses with the different origins between sea surface and atmosphere.

The statistical analysis, consisting of several cruise data including oceanic, atmospheric observations and meteorological elements, is important for the scientific knowledge.

The various case studies, such as anthropogenic pollution and new particle formation, are also interesting topics.

However, the overall format and tone of the paper is not consistent and there are mixed of well-structured paragraph and single sentence statements without clear causal relationships, I would like all co-authors to review the format of the paper so that it is consistent and also the structure in the manuscript is clear.

The English language in general needs to be improved.

Publication can be considered after the following points and issues have been addressed.

We would like to thank the reviewer for his work, which we believe has significantly improved the article. We have done our best to answer his questions and recommendations.

The general comments on the simplification, organization, and presentation of the article are shared by both reviewers.

As a result, we have significantly reduced and simplified the introduction and section 2, and reorganized the presentation of the results to improve the overall consistency and clarity. We have also rewritten the abstract and conclusion.

Section 5 now presents the number of aerosols, the activation diameter, and the hygroscopicity simultaneously. As requested, we have sought to better link our results.

Section 6 uses four specific situations to explain some of the main spatial variations in aerosol properties highlighted in Section 5.

Finally, Section 7 provides a generalization of the results (size and hygroscopicity distribution as a function of wind) according to three categories of air masses.

We believe that this presentation is now clearer and less tedious.

We have done our best to improve our English and avoid simplistic or causally unrelated sentences.

Specific comments:

My main concerns are:

(1) Each topic and result is presented; there are different regions that are relatively clean and mainly from marine sources, while others are affected by pollution from continental sources. The data are interesting, but the description needs to be clear, especially in the abstract, how they are ultimately linked and what the authors want to argue as a conclusion or implication of the results, not separately in individual sections.

You are right, and we took your comment into consideration. We reorganized the whole results part by presenting first NSMPS, NCCN, Dact and κ along the Marion Dufresne track, then we explained the variability of the above parameters focusing on four situations. The third part of the results presents the general behaviour of the aerosol size distribution and κ evolution against wind speed, taking the origin of air mass into account. The abstract has been reworded on this basis.

(2) Some sentences are suggested with kappa values and previous studies but not analysed in this study itself, e.g. processes and chemical compositions, it is desirable to be specified that these are observations in this study or indications from the previous studies or literature.

You are right. In this new version of the manuscript, we presented results from previous studies in the introduction, kept a few elements of the literature in the Results part for comparison with our results, and finally discussed in more detail the similarities and differences between our results and those from the literature in the Conclusion part.

(3) Regarding point (1), although this manuscript is divided into sections, I recommend that the necessary data be reported in detail first and then discussed, or merged into some sections with related topics.

For example, the topic of kappa is related to both the origin of the aerosol mass and the biological activity, so it should be discussed after the description of Figure 8.

You are right. As mentioned in the answer regarding point (1), we gathered the results concerning the spatial distribution of $N_{SMPS}$, $N_{CCN}$, $D_{act}$ and κ in Section 5. Then, we explained the variability of these parameters in Section 6 for four situations that we selected. Finally, as general results, we presented the aerosol size distribution according to the origin of air mass, and the κ evolution against the wind speed and according to the origin of air mass and the nanophytoplankton abundance.

**(Abstract)**

**L8-11:**

**You describe the kappa increase in the subtropical Indian Ocean and the kappa decrease in the southern Indian Ocean, and you suggested the same reasons (with high wind speed in both cases). What is the difference between the kappa values in both cases?**

**Also, if the local emission with wind at the sea surface is large, the highly hygroscopic components (sea salt and sulphate) may also contribute in addition to biological organic components.**

You are right. The results concerning the κ evolution are presented in Section 7.2. We decided to present the results at 0.2 and 0.4 % SS, and at 0.2-0.4 % SS. For the subtropical Indian Ocean group, κ increases with wind speed at both SS. For the southern Indian Ocean group, κ increases with wind speed at 0.2 % SS and is stable at 0.4 % SS. These results can be explained by the low abundance in nanophytoplankton measured for the subtropical Indian Ocean group, which makes κ values closer to NaCl values as the wind speed increases. On the contrary, the nanophytoplankton abundance measured for the southern Indian Ocean group is more elevated and the effect of organic matter in reducing κ values as the wind speed increases is more visible on the smaller aerosols that can act as CCN at 0.4 % SS. This behaviour can be explained because primary organic matter tends to accumulate at the surface of smaller aerosols and counterbalances the emission of NaCl as the wind speed increases.

**L11-13:**

**Last 4 sentences, revise to make the cause and effect clear between sentences.**

**If you want to pick up some case studies, should be written so, with your indication or suggestion of results.**

You are right. The cause and effect between the variation of $N_{SMPS}$, $N_{CCN}$, Dact and κ and the case studies has been accentuated and the sentence has changed to : "Four distinct scenarios are examined to elucidate some of these variations." L6-7.

**(Manuscript)**

**L203-208:**

**Why do you explain the details only for the CCNC here? In this section it is necessary to briefly describe the observation information like other instruments. If you want to describe more, move to the introduction or experimental sessions.**

You are right. The CCNC functionment is now more briefly described in L197-203.

**L284-288 :**

**I cannot follow this part. Do you want to describe the range and the percentage of the total number concentrations? The most important thing is the representative characters rather than those of the small fraction (<10%), and the good explanation is the same as described at L299-300. It is better to use the same format here, i.e. "mostly (97%) xxx... and the rest is xxx...".)**

You are right. This part has been lightened in Section 5 by presenting only a few percentages describing the statistics of each parameter $N_{SMPS}$, $N_{CCN}$, $D_{act}$ and $\kappa$.

**L291:**

**"in this mode": What mode or range do you mean? please clarify.**

You are right, it had to be clarified, we should have written the sentence as follows:

"This explains why the aerosol number concentration is lower in the coarse mode and decreases during transport."

but this sentence has been removed from the revised manuscript.

**L292:**

**"in this mode": What mode or range do you mean? Please clarify.**

You are right, it had to be clarified, we should have written the sentence as follows:

"Thus, the highest aerosol number concentrations in the coarse mode are observed in regions where primary production is more important, as it is the case south of 40 ° S during the three storms identified in Figure 2a."

but this sentence has been removed from the revised manuscript.

**L306-310:**

 "Along this transect, the percentage of aerosols in the free troposphere is higher than..."

Can you add here the percentages of air masses from the free troposphere exactly?

This transect was done between January 21$^{st}$ 2021 and January 23$^{rd}$ 2021. Five days before this period, the percentage of air masses coming from the free troposphere is between 50 % and 75 %. The air mass subsidence into the marine boundary layer is visible two days before they arrive at the vessel's location with a percentage of air mass in the boundary layer between 20 % and 100 %.

This paragraph has however been removed from the new version of the manuscript as we decided to focus on four periods labeled from 1 to 4 on Figure 4, that are analyzed in Section 6.

**L311-315:**

 "Peaks of Na concentration are also visible..."

Can you add here the numerical concentrations, as described in the other three regions (north of Madagascar, south-east coast of Africa, eastern part of the Indian Ocean)?

Thank you for this comment. The numerical concentrations should have been added to the sentence as follows:

"Peaks of N$_{total}$ between 1000 and 1500 cm$^{-3}$ are also visible east of Prince Edward islands (Fig. 4(a), label 1c)."

but this paragraph has been removed from the revised manuscript as we decided to focus on four periods labeled from 1 to 4 on Figure 4, that are analyzed in Section 6.

**L321:**

**weak variation? or low concentration? It's not common as "weak concentrations".**

Thank you for this comment. This is not the appropriate word. The sentence should have been written as follows:

"The coarse mode is associated with concentrations lower than 1 cm⁻³, with most values between 0.1 and 0.6 cm⁻³."

but is no longer in the revised manuscript.

**L373-389:**

**The CCN-derived kappa values might give some indication, but in fact the kappa values in the ambient are determined by the combination of fractions of several chemical components, so you should describe carefully if you don't have chemical composition data or evidence.**

**For example, if kappa were between 0.6 and 1.0 (L386), why do you think the combination of organic matter and sulphate? not the contribution of sea salt?**

**If kappa were around 0.2 (L380), why do you think the organics? Are they transported or local emissions? Because it could be explained by other combinations, such as hydrophobic and hygroscopic components (BC and sulphate). If you have supporting data and reasons (e.g. biological activity), you should describe them here.**

Your assessment is correct. κ can result from various chemical combinations. However, as this area is not influenced by continental pollution and has low CO concentrations (background level), we can assume that BC concentrations must also be very low and have no significant impact on the decrease in κ. We therefore believe that the decrease in κ below the theoretical value for sulfates (0.6) is mainly due to the organic fraction.

However, to determine its origin precisely, in particular whether it is primary or secondary, chemical measurements would need to be carried out and emission fluxes would need to be measured.

In the latest version of the article, we have refrained from speculating on the origin beyond the case studies presented in Section 6.

**L376-377:**

**"which is in line with the previous observations of Na and CCN..." Please add references to the previous studies.**

Thank you for your comment. You are right, the sentence as written can be confusing. L376-377 was written to highlight that the activation diameters calculated along the transect 2 (Fig. 6(a)) is consistent with $N_{SMPS}$ and $N_{CCN}$ measured along this transect and described in Section 4.1 and Section 4.2 of the previous version of the article. But this sentence has been removed from the revised manuscript.

**L404-406:**

**You mentioned the median and interquartile values only in this part, but you discussed the averaged kappa except this sentence.**

**If you want to discuss the median values, it should be added to the values and discussions before the part. If not, I recommend deleting it.**

You are right, it has been removed.

**L463-464:**

**"This result is consistent with the low nanophytoplankton abundance in this group, which makes the κ values closer to NaCl κ"**

**I find the aerosols in the subtropical Indian Ocean had the most hygroscopic values among the three difference groups in Figure 8a, but it is much different from the kappa of NaCl or sea salt (about 1.2-1.3 of the kappa), cannot say "closer". Please revise.**

Thank you for this comment, you are right. The κ values for the subtropical Indian Ocean is effectively much different from the κ of NaCl or the one of pure sea salt.

The sentence has been modified in the paper as follows (L436-438):

"This result is consistent with the lower abundance of nanophytoplankton in this group, which makes the κ values higher than those obtained for continental and southern Indian Ocean air masses, which brings the κ values closer to that of NaCl."

**Figure 8 and L461-476:**

**I cannot follow the explanation of Figure 8.**

**The organic fraction tends to be high in the fine particles especially high biological activity period and they can act as surfactants and affect to the CCN activation ability, as you suggested.**

**However, I think that the organic fractions at 76 nm and 105 nm, around the two activation diameter at 0.2% and 0.4%, is not so much different, indeed mass fraction of 60-125 nm was presented in the same size range in O'Dowd et al., 2004.**

**Recently, primary bioaerosol concentrations by local emission at remote oceans are well correlated with wind speed, corresponding to the wind-driven upwelling from the sea surface to the atmosphere (Kawana et al., 2021), and also discuss the marine (primary) bioaerosol formation for the submicron particles (Santander et al., 2021) by fluorescence methods in seawater and ambient samples.**

**Do you have any insights to explain for your results?**

**Can you also look at the kappa variability at 0.2% SS?**

You are right. The mass fraction of 60-125 nm is not much different in O'Dowd et al., 2004.

Therefore, we believe that primary organic matter compensates for the increase in K through the emission of NaCl, regardless of aerosol size. In our opinion, the difference in K evolution observed between the two supersaturation is due to the accumulation of surfactants on the surface, which reduces the hygroscopicity of the smallest aerosols more strongly (due to their lower surface/volume ratio): for the same organic/NaCl mass ratio at emission, the surface concentration of surfactants will decrease with size.

In the revised manuscript, the κ variability at 0.2 % SS has also been presented. For the southern Indian Ocean group, a clear increase in the hygroscopicity is visible as the wind speed increases. This increase in K is greater than in the South Indian Ocean for the same wind ranges.

Regarding the bioaerosols (e.g., bacteria and viruses) identified by Kawana et al. (2021), to our knowledge, they are larger than the submicron aerosols (less than 300 μm) identified by the SMPS. We believe this primary organic matter originates from sugars and lipids produced by phytoplankton and bacteria.

However, the measurements from the MAP-IO program do not allow us to identify this organic matter.

**L562:**

**"κ values of organic species (0.22-0.24) are observed at 0.4% SS during storm events".**

**You derived the kappa values, but can you say these values is for organic species? or is this based on the indication from the literature?**

Thank you for your comment. The kappa values have effectively been derived from the SMPS and CCN measurements. The kappa values of organic species have been reported in previous studies and are considered to fall between 0.01 and 0.4 (Petters and Kreidenweis, 2007). It is assumed that these values are linked to the presence of organics due to the distance from other sources of hydrophobic aerosols (e.g. pollution), because the sea state and meteorological conditions are conducive to the emission of primary marine aerosols, and because phytoplankton abundance is elevated in this region.

Another indicator is CO concentrations, which are within background values. This is thought to be a good marker for the absence of significant BC concentrations.

This sentence is no longer in the revised manuscript.

**Technical comments:**

**L3 (and in manuscript):**

**Total number of aerosols: Na should be changed to $N_{total}$ or $N_{aero}$ etc, Na leads to confusing as sodium.**

It has been corrected. Na has been changed to $N_{SMPS}$.

**L177:**

**Use the same format in the paragraph, although you have used both number (i.e. 19 instruments) and English (three OPC-N3).**

**e.g. Nineteen instruments for gas and particles and remote sensing...?**

It has been corrected (L180).

**L178:**

**Among the nineteen instruments, seven instruments...?**

It has been corrected (L180).

**L208-210 (and in manuscript):**

**Change to the subscript, e.g. $NO_x$, $O_3$, …**

It has been corrected.

**L259:**

**FLEXible PARTicle Model (FLEXPART)... the word should be spelled out the first time it appears.**

It has been corrected (L258).

**L339 (and in manuscript):**

**Why don't you spell out "CCN number concentrations" as $N_{CCN}$, like $N_{total}$? It's a frequency and usually $N_{CCN}$ is used.**

It has been corrected.

**Table 2:**

**"D = 1.7 nm" in coarse mode over the subtropical Indian Ocean -> "D = 1.7 um"**

It has been corrected.

**L438:**

**"Larger N" -> Larger Total Number Concentrations (N)**

This sentence is no longer in the revised manuscript.

**L556-557:**

**CCN-100 measurements -> CCN measurements**

This sentence is no longer in the revised manuscript.

**Figure captions:**

**Revise to use the same format with figure description and caption, i.e. (a) wind speed, (b) wave height, ... in Figure 2 (and other figures).**

It has been corrected.

References:

1) Kawana, K., Matsumoto, K., Taketani, F., Miyakawa, T., and Kanaya, Y.:

Fluorescent biological aerosol particles over the central Pacific Ocean: covariation with ocean surface biological activity indicators,

Atmos. Chem. Phys., 21, 15969–15983, https://doi.org/10.5194/acp-21-15969-2021, 2021.

2) O'Dowd, C. D., Facchini, M. C., Cavalli, F., Ceburnis, D., Mircea, M., Decesari, S., Fuzzi, S., Yoon, Y. J., and Putaud, J.-P.:

Biogenically driven organic contribution to marine aerosol, Nature, 431, 676–680, https://doi.org/10.1038/nature02959, 2004.

3) Santander, M. V., Mitts, B. A., Pendergraft, M. A., Dinasquet, J., Lee, C., Moore, A. N., Cancelada, L. B., Kimble, K. A., Malfatti, F., and Prather, K. A.:

Tandem Fluorescence Measurements of Organic Matter and Bacteria Released in Sea Spray Aerosols,

Envrion. Sci. Technol., 55, 5171–5179, 2021.

**Referee 2:**

**General comments:**

**Dournaux et al. present observations of aerosol number size distribution, cloud condensation nuclei, activation diameters, and hygroscopicity from six different shipborne marine campaigns in the southwest Indian Ocean.**

**The presented results are interesting and offer valuable insight into marine aerosols. The scientific basis of the study, including the campaigns and the analysis, are mainly solid and there are no major scientific issues. However, the manuscript is noticeably unevenly written, and has issues both in the grammar of the writing and the structure of the text itself. In addition, I find the discussion on the manuscript lacking. The authors fail to properly discuss the implications and argue the importance of their results. As it is, the manuscript reads more like a measurement report.**

**If the authors go through the manuscript carefully and address these issues presented, I believe this manuscript by Dournaux et al. has the potential to be published, and be published as a research article.**

We would like to thank the reviewer for his work, which we believe has significantly improved the article. We have done our best to answer his questions and recommendations.

The general comments on the simplification, organization, and presentation of the article are shared by both reviewers.

As a result, we have significantly reduced and simplified the introduction and section 2, and reorganized the presentation of the results to improve the overall consistency and clarity. We have also rewritten the abstract and conclusion.

Section 5 now presents the number of aerosols, the activation diameter, and the hygroscopicity simultaneously. As requested, we have sought to better link our results.

Section 6 uses four specific situations to explain some of the main spatial variations in aerosol properties highlighted in Section 5.

Finally, Section 7 provides a generalization of the results (size and hygroscopicity distribution as a function of wind) according to three categories of air masses.

We believe that this presentation is now clearer and less tedious.

We have done our best to improve our English and avoid simplistic or causally unrelated sentences.

**Specific comments**

*Abstract:*

**L11-L13: Poorly written, rewrite this as e.g., "High aerosol concentration events are presented, and include: pollution related air masses…"**

Thank you. We have decided to rewrite the abstract and this sentence no longer exists.

*Introduction*

**The introduction is a bit hard to read and has some structural and grammatical issues. Many of the paragraphs are too long, and therefore very heavy to read. The text in general lacks flow. Note that the readers typically pay most attention to the start and end of paragraphs, while skimming through the middle.**

**In addition, the main objectives of the study are not clearly stated.**

Thank you for your comment, the introduction has been revised and we hope that the new version is easier to read. The main objectives of the study have also been reworded and stated L108-112.

**L16: "Among them, marine aerosols...". It would put these values more into perspective, if it were also stated how large of a fraction this is relative to the total global aerosol emissions.**

You are right, adding the fraction of marine aerosols emitted relative to the global aerosol emission would have put these values more into perspective, however this fraction is not specified, or only for certain types of marine aerosols, in the articles read by the authors.

**L17: Please define "marine aerosol" as aerosol including all types of particles found over the oceans, regardless of point of origin.**

Thank you for your comment. The definition has been added to the introduction L29-30 as follows:

"Marine aerosols are defined as aerosols comprising all types of particles found over the oceans, regardless of their point of origin."

**L37: The paragraph starting here lasts for one and half pages and is difficult to read. Revise the structure so that it can be divided into multiple shorter paragraphs.**

You are right. This paragraph has been reduced and as you suggested, we revised its structure dividing it into multiple shorter paragraphs L49-89.

**L111: "This paper is organized as follows" could start a new paragraph. I would also add the objectives/aims of the study to this paragraph, before the structure of the paper is described.**

Thank you for your comment. We started a new paragraph for the presentation of the sections of the manuscript L114.

We added the objectives of the study in the previous paragraph (L109-113) to enhance the clarity.

*Campaigns overview and in situ conditions observed*

**2.2 Atmospheric and oceanic conditions observed: the whole section is one, long paragraph. Revising the structure would make it both easier to read and present the relevant information in a more clear manner.**

You are right. As you suggested, the structure of this section has been reduced to present the relevant information.

**L183: "The distance between the inlets and the instruments (8 m) was carefully chosen" Can you give a little bit more details about this as the distance between the inlets and the instruments is quite long and could result in a significant amount of diffusion and deposition losses.**

The instruments are located next to the wheelhouse in an accessible and secure area of the ship. A compromise was made between the distance and the acquisition area. 8m is a considerable distance, but should not be too limiting for submicronic particles. For the coarse mode, investigations will have to be carried out to qualify and determine the cut-off size for the validity of the aerosol size. This distance is also the reason why we chose to install OPC-N3 (low-cost) sensors directly on the ship's deck for coarse aerosol measurement. Overall, we make little use of these measurements in the paper.

**L189: Can you discuss a bit on the impact of measuring the dry particles on your results?**

According to the recommendation of the Center for Aerosol In-Situ Measurements (IR ACTRIS): "Aerosol in-situ measurements should be done at a relative humidity lower than 40 %. This is necessary to obtain comparable data, independent of the hygroscopic behavior of the aerosol particles. "

https://www.actris.eu/news-events/news/ecac-preliminary-recommendations-aerosol-situ-sampling-measurements-and-analysis

**L211: Can you specify, which variables have which time step?**

You are right, this sentence needed to be rewritten to be clearer.

The Vaisala and Mercury weather stations measure the same meteorological parameters, as to say, wind speed, wind direction, air temperature, air pressure, and humidity. The time step of the Vaisala station is 5 seconds and the time step of the Mercury station is 1 minute.

The sentence has been changed to : "Along with aerosol measurements, wind speed, and wind direction (m s$^{-1}$ and °), air temperature (°C), pressure (hPa), and humidity (%) are recorded by the Vaisala meteorological station with a sampling time step of 5 seconds, and by the Mercury meteorological station with a sampling time step of 1 minute."

*Spatial and temporal variability of marine aerosols properties*

**As a general comment for this section (and the following section 6) and its subsections, there is a lot of results and observations. However, I would like more discussion and emphasis on their relevance. In addition, at some parts it is hard to remember what has been already discussed before in this section and how it is related to what is discussed later on.**

You are right. Your comment is in line with the one of the 1$^{st}$ reviewer. Following your suggestion, the structure of the manuscript has been revised to better highlight the different results and discuss them together.

We gathered the results concerning the spatial distribution of $N_{SMPS}$, $N_{CCN}$, $D_{act}$ and κ in Section 5. Then, we explained the variability of these parameters in Section 6 for four situations that we selected. As general results, we presented the aerosol size distribution according to the origin of air mass, and the κ evolution against the wind speed and according to the origin of air mass and the nanophytoplankton abundance. Finally, we discussed our results with respect to the literature in the Conclusion part.

We hope that this new structure will be clearer and more concise.

**L291: Please specify, which mode.**

You are right, it had to be clarified. The mode discussed here was the coarse mode.

But, this sentence is no longer in the new version of the manuscript.

**L293: "The difference in the location of high and low aerosol concentrations reflects…" This sentence would make more sense placed before the discussion on the processes starts, i.e., before "Sedimentation and below-cloud scavenging…"**

Thank you for your comment. In the revised manuscript this sentence has been removed from the text.

**L296: "… nucleation of sulfuric acid" To my knowledge, this is not fully accurate as a) the nucleation mechanism in varying marine environments is still poorly understood, b) both iodic acid and methane sulfonic acid have been in previous studies been identified as participating in the nucleation process in marine environments.**

You are right. Both iodic acid and methane sulfonic acid have been identified in previous studies as participating in the nucleation process in marine environments. Therefore, we cannot say that sulfuric acid only is implicated in the nucleation process occurring in marine environments.

This sentence has been removed from the revised manuscript.

**L308-L309: "Along this transect, the percentage of aerosols in the free troposphere..." I may have missed something, but what is the basis for this? Where does the information on the percentage of aerosols in free troposphere come from?**

You are right. This transect was done between January 21$^{st}$ 2021 and January 23$^{rd}$ 2021. Five days before this period, the percentage of air masses coming from the free troposphere is between 50 % and 75 %. The air mass subsidence into the marine boundary layer is visible two days before they arrive at the vessel's location with a percentage of air mass in the boundary layer between 20 % and 100 %.

This paragraph has however been removed from the new version of the manuscript as we decided to focus on four periods labeled from 1 to 4 on Figure 4, that are analyzed in Section 6.

**L322: Please specify why these values suggest that the air masses have aged.**

This paragraph, as it appeared in the first version of the manuscript, has been rewritten. At the stage of discussion in section 5, it is not possible to indicate that the air mass is old. This will be shown with the SMPS spectrum during the situation analysis in the new section 6.

**L339: CN is defined, but not used aside from defining the ratio CCN/CN, which is not actually discussed in this manuscript.**

You are right. We decided to use N$_{SMPS}$ in the whole manuscript instead of CN to refer to the total number of aerosols measured between 3 nm and 350 nm by the SMPS.

The ratio CCN/CN has been briefly discussed in the Conclusion to compare continental/polluted areas to less continental/polluted areas, using the term "N$_{CCN}$/N$_{SMPS}$" (L457).

*Aerosols size distribution based on their air mass origin*

**6.1 Size distribution of marine aerosols: This whole subsection is one, too long, paragraph. Please revise. In addition, there should be more discussion on the relevance of the results.**

You are right. This subsection, now Subsection 7.1 has been simplified to the more relevant information.

**L438: "Plus, the larger N in the Aitken mode of the continental air masses implies that there will be a more important coagulation process...". I do not follow the line of reasoning here, please clarify.**

You are right. This sentence had to be clarified and has been modified to:

"The Aitken mode of the continental group has the largest average diameter, and there is no clear separation in terms of number concentration between this mode and the accumulation mode. This distinction is particularly noticeable when compared to air masses from the subtropical Indian Ocean group. This can be explained by the pollutant load (gases and aerosols) of the continental group, which favors growth through condensation and coagulation."

**L443-L450: Structurally, it would work better if throughout the section you first state what you have observed and then discuss it and possibly refer to a previous study, and then tell what else you have observed, discuss it and refer again to another study. Now the last 10 lines of this subsection are spent on referring to other studies without any discussion on how they relate to your study, while they should be spent on emphasizing your own results and their implications.**

You are right. We revised the structure of this section. We described and discussed the main differences between the aerosol size distribution related to different air mass origin. We then referred to other studies in the Conclusion and discussed how our results were different from those of these studies.

*Focus on particular events*

**I would consider showing the same information, at least as supplementary material, for each of the cases.**

**Here, as in the previous sections, I wish there was more discussion of the results and not just reporting what was observed during these different cases. Do we learn anything new from these cases? How do these cases connect to previous studies? These cases are interesting, and it should be made more clear why they are interesting.**

Thank you for your comment. We have attempted to provide more detail on what each case study contributes and to link them to the observations in Section 5.

We use them as examples to explain the variability in observations of aerosol number, CCN, and the hygroscopicity parameter. In accordance with your suggestion, we have improved the physical explanations by detailing the link between the origin of the air mass and observations.

For each of the cases we decided to show the same information in the first four panels of the figures:

-1st panel presents the Marion Dufresne location and satellite observations of precipitation or cloud cover, and back trajectories over the period along which are shown different information according to the case study (altitude of air mass or 10 m wind speed)

-2nd panel presents the temporal evolution of the aerosol size distribution measured by the SMPS along with the activation diameter at 0.2 and 0.4 % SS

-3rd panel presents the aerosol concentration (from 3 to 350 nm, NSMPS) and cloud condensation nuclei (NCCN) at 0.2 and 0.4 % SS

-4th panel presents the κ evolution at 0.2 and 0.4 % SS

And 5th panel presents information specific to the case study (mixed boundary layer thickness and CO concentration for the polluted case, wind speed and wave height for the storm and rain cases)

These four cases are presented in Section 5 and analyzed in Section 6.

*Conclusions*

**There's quite a lot of detail, and the conclusions might be clearer if it was slightly more condense.**

The conclusion has been simplified and improved.

**L555: Please be more exact and clarify what are the weather influences behind the highest concentrations.**

You are right. The weather influences behind the concentrations observed have been clarified.

*Technical comments*

**L4, L42, rest of the manuscript: Please write Na using subscript and italics.**

It has been corrected. Na has been changed to $N_{SMPS}$.

**L29: "Low clouds, such as"**

It has been corrected (L41).

**L30: "has" → "have"**

It has been corrected (L42).

**L31: "changes, which"**

It has been corrected (L43).

**L35-L36: "the lowest", "the greatest"**

It has been corrected (L47-48).

**L85: > 1**

It has been corrected and changed to "~1" (L86).

**L96: "to the authors knowledge" should come before "over"**

It has been corrected (L98).

**L109: "composition, which"**

This is no longer in the revised manuscript.

**L498: "clearly indicates" is repeated twice on sentences following each other, please consider rephrasing.**

It has been corrected. The sentence "Back-trajectory analysis of Figure 9 (a,b,c,d) clearly indicates the passage of the air mass over the urbanized region of Majunga, located in north-western Madagascar" has changed to "The backtrajectory analysis clearly shows the

passage of the air mass at about 800 m a.g.l. over the urbanized region of Majunga, northwest Madagascar."

**Citation: https://doi.org/10.5194/egusphere-2024-3747-RC2**

---

## Author Response (AR2)

**Referee 1:**

Thank you for addressing the comments, the revised manuscript is well improved to understand and read, and story structure/quality is better.
I want to add just one comment here;
Line 6 in abstract, you mentioned the 4 distinct scenarios as potential variability. It is a bit difficult to follow the length (I guess you describe it in lines 8-15?), so I suggest that you describe shorter sentence to be clear the 4 distinct scenarios with the characterization of the air mass/regional differences (based on Fig. 5 and Fig. A1), if my understanding is correct. i.e., (1) the predominance of polluted air mass in the Mozambique Channel, with the weakly hydrophilic aerosols,... (2) the predominance of pristine marine in xx, ... (3) with precipitation/strom events in xx, ...(4) with NPF events in xx, etc.

Thank you for your comment. We have modified the abstract (L8-15) as suggested to describe more clearly and concisely the distinct scenarios. Thank you for your help.

The paragraph has been modified in the paper as follows (L6-11):

"Four distinct scenarios are examined to elucidate some of these variations. (1) the predominance of pristine air mass in the eastern regions of the subtropical Indian Ocean, with highly variable κ values sensitive to the low aerosol concentration measured in this area (2) the predominance of polluted air mass in the Mozambique Channel, with weakly hydrophilic aerosols (3) a precipitation and storm event in the southern Indian Ocean, with highly variable κ values (4) a NPF event in the open ocean, with an increase in κ values as the new particles formed grow to Aitken mode particles."

**Editor:**

Public justification (visible to the public if the article is accepted and published):
Editor comments for Dournaux et al. paper entitled "Origin, size distribution and hygroscopic properties of marine aerosols in the south-western Indian Ocean: report of 6 campaigns of shipborne observations" submitted for Atmos. Chem. Phys.

Thank you for responding to the reviewer comments. There are few more comments by the reviewer 1. Please respond to them.

In addition to the reviewer comments, I have few editorial comments regarding the structure and wording in the paper.

The title should reflect a bit better the content of the paper. I suggest to change the word "report" to "results" in this way: "Origin, size distribution and hygroscopic properties of marine aerosols in the south-western Indian Ocean: results of 6 campaigns of shipborne observations".

The name of the sections and their order should be improved.

I provide a suggestion below. The content can stay almost the same. In the Results and Discussion section I suggest to first present the general results (properties, size distribution, relationships) and only then present the case studies. The figure numbering would change as the consequence as well as the order of paragraphs in the conclusions section.

2. Overview of the campaigns and observations
2.1 Campaigns
2.2 Geographical and meteorological context
2.3 Instrumentation

3. Data processing
3.1 Data filtering
3.2 Activation diameter and hygroscopicity parameter
3.3 Air mass classification

4. Results and discussion
4.1 Spatial and temporal variability of marine aerosols properties
4.2 Size distribution of marine aerosol particles
4.3 Relationship between marine aerosol hygroscopicity, wind speed, and nanophytoplankton abundance
4.4 Case studies
(please write a short justification of the selected case studies, 1-2 paragraphs)
4.4.1 Pristine case (OBSAUSTRAL)
4.4.2 Polluted case (SCRATCH)
4.4.3 Storm and rain case (SWINGS)
4.4.4 Nucleation case (OP3)

5. Conclusions

End of the conclusions, I suggest to edit the last paragraph (integrate with the previous paragraph):

"This paper highlights the need to incorporate the variability of marine aerosol CCN properties into meteorological models, emphasizing the complexity of their characterization due to various coupled processes involving emissions, transport, aging, and chemical composition."

Please provide few sentences of the impacts that could be reached after doing this.

Figure 1. Please explain AC, SIC, NEMC, SEMC, SEC acronyms in the figure caption.

Yours,
Tuukka Petäjä

Thank you for your comments. As suggested, we have modified the title of the manuscript, the titles of the sections and subsections. We find them clearer and more concise. We have explained the acronyms in the caption of Figure 1. We have also edited the last paragraph of the conclusion, added a concluding sentence, and modified the abstract as recommended. Thank you for your help.

The reorganization of the outline you propose is the same as the one we had initially proposed in the first submission. However, one of the main comments made by both reviewers was to reorganize this outline to better explain the variability of the observations made during the campaigns. We therefore proposed that the case studies (section 5) be used to understand the spatial and temporal variability and heterogeneity of the observations in Figure 4.

Section 7 is then a generalization of the observations by region, but does not include the temporal dimension of the observations along the ship's route.
Objectively, we agree with the reviewers and believe that this new organization is clearer in the presentation of the results.
We therefore prefer to keep this outline.

We also corrected the sentence L390 as there was a mistake:

"The $\kappa$ at 0.2% of SS increases at 0.46."

Here we are describing the increase of $\kappa$ at 0.2% SS and not at 0.4% SS.